# Dyslipidemia: A Narrative Review on Pharmacotherapy

**DOI:** 10.3390/ph17030289

**Published:** 2024-02-23

**Authors:** Lucas Lentini Herling de Oliveira, Arthur Cicupira Rodrigues de Assis, Viviane Zorzanelli Rocha Giraldez, Thiago Luis Scudeler, Paulo Rogério Soares

**Affiliations:** Instituto do Coração (InCor), Hospital das Clínicas da Faculdade de Medicina da Universidade de São Paulo, Av Dr Enéas de Carvalho Aguiar, 44, Cerqueira César 05403-000, Brazil; lucas.lentini@hc.fm.usp.br (L.L.H.d.O.); arthurcicupira@usp.br (A.C.R.d.A.); vzrocha@hotmail.com (V.Z.R.G.); paulo.soares@hc.fm.usp.br (P.R.S.)

**Keywords:** dyslipidemias, cholesterol, statins, gene therapy

## Abstract

Dyslipidemia plays a fundamental role in the development and progression of atherosclerosis. Current guidelines for treating dyslipidemia focus on low-density lipoprotein–cholesterol (LDL-C). Despite advances in the pharmacotherapy of atherosclerosis, the most successful agents used to treat this disease—statins—remain insufficient in the primary or secondary prevention of acute myocardial infarction. Advancing therapy for hypercholesterolemia with emerging new drugs, either as monotherapy or in combination, is expected to improve cardiovascular outcomes. An emerging field in dyslipidemia pharmacotherapy is research on genetic therapies and genetic modulation. Understanding the genetic mechanisms underlying lipid alterations may lead to the development of personalized treatments that directly target the genetic causes of dyslipidemia. RNA messenger (mRNA)-based therapies are also being explored, offering the ability to modulate gene expression to normalize lipid levels. Furthermore, nanotechnology raises new possibilities in drug delivery for treating dyslipidemia. Controlled-release systems, nanoparticles, and liposomes can enhance the effectiveness and safety of medications by providing more precise and sustained release. This narrative review summarizes current and emerging therapies for the management of patients with dyslipidemia.

## 1. Introduction

Dyslipidemia is one of the most important risk factors for atherosclerotic cardiovascular disease (ASCVD). In 2017, high non-HDL cholesterol was responsible for an estimated 3.9 million deaths worldwide [1]. Therefore, lipid-lowering therapies, especially statins, have been shown to be cost-effective or cost-saving, particularly in people with a high CV disease risk [2,3,4,5]. Although not the focus of this review, it is worth mentioning that effective community-based prevention strategies promoting lifestyle modification (e.g., dietary improvement and regular physical activity) are also needed to control dyslipidemia.

Cholesterol is a hydrophobic molecule, insoluble in plasma, with several vital functions in our body, such as the production of hormones and the formation of cell membranes. Since the discovery of cholesterol at the end of the 18th century, when it was isolated from gallstones, to its association with atherosclerosis, vast knowledge has been accumulated about the molecule, its metabolism, and its role in atherosclerosis.

Due to its insoluble nature, cholesterol is transported in plasma through lipoproteins, generally spherical structures made up internally of nonpolar lipids, such as cholesterol esters and triglycerides, and externally by polar lipids such as phospholipids, apolipoprotein, and free cholesterol [6]. On its surface, we observe the presence of apolipoproteins (apo), which are fundamental structures for the signaling, transport, and binding of lipoproteins to receptors. Due to their amphiphilic nature (membrane-forming molecules), they are crucial in the stability and function of lipoproteins.

Evidence from epidemiological and clinical studies supports a key role of circulating LDL-C and other apolipoprotein B (apoB)-containing lipoproteins in atherogenesis. Although the benefits of lipid lowering are well established in high-risk individuals, a number of trials show that the benefits extend to lower-risk individuals as well. Knowledge of cholesterol metabolism is essential for understanding dyslipidemia and the drugs used in its treatment.

Although statins remain the first line of pharmacotherapy, novel lipid-lowering therapies are currently available, such as PCSK9 inhibitors; gene therapy, including small interfering RNAs (inclisiran); ANGPTL3 inhibitors (evinacumab); CRISPR/Cas9, antisense Oligonucleotides (mipomersen); apoB and MTP Inhibitors; and, finally, vaccines against PCSK9 and targeted nanotherapy.

The aim of this article was to review currently available therapies and emerging therapeutic agents for the management of patients with dyslipidemia, in light of recent evidence and guideline recommendations.

## 2. Methods

This narrative review was conducted by searching Pubmed, Scopus, and Web of Science using the following terms: hydroxymethylglutaryl-CoA (HMG-CoA) reductase inhibitors, proprotein convertase subtilisin/kexin type 9 (PCSK9) inhibitors, ezetimibe, bempedoic acid, inclisiran, angiopoietin-like 3 (ANGPTL3) inhibitors, inhibitors of apoB, Lp(a)-targeted therapies, microsomal triglyceride transfer protein (MTP) inhibitors, the inducible degrader of LDL receptors (IDOL), bile acid-binding resins, nicotinic acid, fibric acid derivatives, cholesteryl ester transfer protein (CETP) inhibitors, gene therapy, vaccines against PCSK9, plasmapheresis, and targeted nanotherapy (Figure 1). The primary lipid-lowering drugs recommended by the European and American dyslipidemia guidelines are included in this review.

## 3. HMG-CoA Reductase Inhibitors

HMG-CoA reductase inhibitors, or statins, are the cornerstone of LDL-C-lowering therapy and are currently recommended as first-line therapy for the secondary prevention of atherosclerotic disease and for primary prevention in at-risk patients [7,8]. These drugs achieve LDL-C lowering by reducing cholesterol synthesis in the liver, ultimately leading to an increase in LDL receptors (LDLRs) in hepatocytes. The enhanced expression of LDLRs on the surface of hepatocytes results in an increased removal of LDL-C from circulation [7].

Statins have a relatively predictable effect on LDL-C. Low-intensity statins (simvastatin 10 mg, pravastatin 10–20 mg) reduce LDL-C by less than 30%, moderate-intensity statins (simvastatin 20–40 mg, atorvastatin 10–20 mg, rosuvastatin 5–10 mg, pravastatin 40–80 mg) reduce LDL-C by 30–50%, and high-intensity statins (atorvastatin 40–80 mg or rosuvastatin 20–40 mg) reduce LDL-C by at least 50%. These values reflect population averages and may not be entirely applicable to individual patients [7,8].

Beyond LDL-C effects, statins also produce modest reductions in triglyceride levels and may lead to discrete increases in HDL-C, usually with a neutral effect on lipoprotein(a) [Lp(a)]. The classical “pleiotropic effects” of statins traditionally refer to the potential anti-inflammatory and antioxidant effects of the drug [7].

Regarding clinical applications for statins, this review is divided into primary prevention, secondary prevention, and special groups, such as heart failure (HF) and chronic kidney disease (CKD) patients. The main trials in these categories are summarized in Table 1, Table 2 and Table 3.

### 3.1. Primary Prevention

In a primary prevention setting, any intervention aimed at reducing challenging outcomes such as mortality or myocardial infarction (MI) must involve large and/or long trials with sufficient power to detect differences in the inherently low event rate when compared to secondary prevention trials. Furthermore, the highest degree of scrutiny and critical reasoning is necessary before recommending an intervention to asymptomatic individuals, since its benefits tend to manifest in the long term, while unaccounted adverse effects may arise from any kind of intervention.

One of the first and most relevant trials that rose to this challenge was the West of Scotland Coronary Prevention Study Group (WOSCOPS) trial [9]. In this study, pravastatin 40 mg was tested against placebo in patients with elevated LDL-C levels and no documented coronary artery disease (CAD). Over the 4.9 years of follow-up, LDL-C levels were reduced by an average of 26%, and patients in the pravastatin group had lower rates of MI and coronary-heart-disease-related death.

Subsequently, the Air Force/Texas Coronary Atherosclerosis Prevention Study (AFCAPS/TexCAPS) evaluated the effect of cholesterol lowering with lovastatin in patients with moderately elevated lipid levels without clinically evident ASCVD [10]. Lovastatin reduced the risk of the primary outcome of MI, unstable angina (UA), or sudden cardiac death (SCD). However, due to the low cardiovascular (CV) risk of the enrolled patients, the absolute risk reduction (ARR) of 0.2% per year was much smaller than demonstrated in previous trials (number needed to treat [NNT] of 86). Furthermore, the study was stopped early for efficacy. Statistical simulations, however, suggest that truncated studies overestimate the magnitude of benefit of the treatment being evaluated by up to 29% [11].

Two other important studies that showed the CV benefits of statins in patients without documented ASCVD were ASCOT-LLA and MEGA.

The Anglo-Scandinavian Cardiac Outcomes Trial—Lipid-Lowering Arm (ASCOT-LLA) trial showed that, among patients with hypertension and relatively low cholesterol, treatment with atorvastatin was associated with a reduction in the primary endpoint of MI or coronary death at a 3-year follow-up [12].

The Management of Elevated Cholesterol in the Primary Prevention Group of Adult Japanese (MEGA) trial showed that treatment with pravastatin in addition to diet modification was associated with a reduction in coronary heart disease events compared with diet modification alone at a mean 5.3-year follow-up [13].

Despite the evidence provided by the WOSCOPS study, concerns have arisen regarding patients with lower LDL-C levels but with an estimated risk of ASCVD. Thus, markers capable of detecting patients who may benefit from statin therapy for primary prevention have been investigated. The Justification for the Use of Statins in Prevention: an Intervention Trial Evaluating Rosuvastatin (JUPITER) trial compared rosuvastatin vs. placebo in patients with LDL-C < 130 mg/dL, no known ASCVD and high-sensitivity C-reactive protein (hsCRP) levels of 2.0 mg per liter or higher [14]. The study showed a reduction in the primary composite endpoint (MI, stroke, arterial revascularization, hospitalization for UA, or CV death) [HR 0.56, 95% CI 0.46 to 0.69, ARR of 0.59% per year for the primary endpoint, NNT 169]. For coronary events, including fatal or non-fatal myocardial infarction, 500 persons need to be treated for one year to prevent one event.

Finally, the Heart Outcomes Prevention Evaluation (HOPE)-3 trial showed that, in patients with intermediate risk (estimated annual rate of major adverse cardiovascular events [MACEs] ~1%), rosuvastatin resulted a reduction in the composite coprimary endpoint of CV death, MI, or stroke compared to placebo (HR 0.76, 95% CI 0.64 to 0.91, NNT of 91) [15].

Despite the cardiovascular benefits suggested by the aforementioned trials, it is important to not overlook clinical reasoning and to recognize the magnitude of the findings. Due to the inherently low incidence of events in the primary prevention population, it is important to acknowledge that the clinical benefit is marginal during the follow-up period of the trials, resulting in high NNTs. Additionally, older patients comprise the majority of these study populations. The long-term benefits are likely greater than those found in the aforementioned trials, and lifelong LDL-C-lowering therapies might have a more significant impact when considering younger patients with high LDL-C levels or higher than average CV risk factors.

**Table 1 pharmaceuticals-17-00289-t001:** Primary prevention trials. Legend: CAD: coronary artery disease; CV: cardiovascular; hsCRP: high sensitivity C-reactive protein; MI: myocardial infarction; UA: unstable angina; TC: total cholesterol.

Study	Sample Size	Characteristics of Patients	Comparison Groups	Follow-Up	LDL-C Reduction	CV Effects
WOSCOPS (1995) [9]	6595	TC > 252 mg/dL	Pravastatin 40 mg vs. placebo	4.9 years	26%	Reduction in MI or coronary death (HR 0.69, 95% CI 0.57 to 0.83, NNT 111)
AFCAPS/TexCAPS (1998) [10]	6605	LDL-C 130–180 mg/dL	Lovastatin 20–40 mg vs. placebo	5.3 years	25%	Reduction in coronary events (HR 0.63, 95% CI 0.50 to 0.79, NNT 86)
ASCOT-LLA (2003) [12]	10,305	Hypertension and CV risk factors	Atorvastatin 10 mg vs. placebo	3.3 years	35%	Reduction in MI or coronary death (HR 0.64, 95% CI 0.50 to 0.83, NNT 83)
MEGA (2006) [13]	7832	TC 220–270 mg/dL	Pravastatin 10 mg vs. placebo	5.3 years	18%	Reduction in CAD (HR 0.67, 95% CI 0.49 to 0.91, NNT 119)
JUPITER (2008) [14]	17,802	LDL-C < 130 mg/dL + hsCRP ≥ 2 mg/L	Rosuvastatin 20 mg vs. placebo	1.9 years	50%	Reduction in CV death, MI, stroke, arterial revascularization, or UA hospitalization (HR 0.56, 95% CI 0.46 to 0.69, NNT 169)
HOPE-3 (2016) [15]	12,705	Intermediate CV risk (CV event rate 1%/year)	Rosuvastatin 10 mg vs. placebo	5.6 years	26.5%	Reduction in CV death, MI, or stroke (HR 0.76, 95% CI 0.64 to 91, NNT 91)

### 3.2. Secondary Prevention

The consistent relative risk reduction (RRR) of MACEs points towards a robust relationship between statin use and a lower incidence of events. However, the clinical significance of such reduction will depend on the absolute rate of events, with more evident benefit observed in patients at the highest risk of MACEs. This is evident in secondary prevention clinical studies.

The Scandinavian Simvastatin Survival Study (4S) trial randomized 4444 patients with coronary artery disease (CAD), defined by prior MI or angina, to simvastatin or placebo [16]. The trial was stopped early due to an ARR of 3.3% in all-cause mortality with simvastatin (11.5% vs. 8.2%; *p* = 0.0003; NNT 30). In addition to being a truncated study, the low rate of aspirin use among the 4S trial population (~37%) draws attention. Possibly, the magnitude of the benefit would be attenuated with widespread aspirin use.

The Cholesterol and Recurrent Events (CARE) trial confirmed a reduction in coronary events in patients with previous MI, even in a group with lower total cholesterol levels (<240 mg/dL, mean LDL-C of 139) [17] Similarly, the Long-term Intervention with Pravastatin in Ischaemic Disease (LIPID) trial showed a reduction in coronary death in patients with previous MI or hospitalization for UA (NNT of 53) [18]. The Heart Protection Study (HPS) showed a reduction in all-cause mortality, driven by vascular causes (7.6% vs. 9.1%, HR 0.83, 95% CI 0.75 to 0.91, NNT 67) in high-risk patients—65% with previous CAD [19]. On the other hand, the Fluvastatin On Risk Diminishing after Acute Myocardial Infarction (FLORIDA) trial showed that fluvastatin did not reduce coronary events in post-MI patients [20]. However, the trial was underpowered, and a post hoc analysis revealed a trend towards a reduction in the primary endpoint in patients with pronounced ischemia at the trial onset [21].

So far, the relationship between intervention and outcome seems to be established, with greater benefits naturally seen in patients at a higher baseline risk. However, a new question has arisen: What is the best statin regimen for reducing cardiovascular outcomes?

The Pravastatin or Atorvastatin Evaluation and Infection Therapy—Thrombolysis in Myocardial Infarction 22 (PROVE IT-TIMI 22) trial tested different intensity regimens in patients in the acute phase of an MI [22]. Among those up to 10 days after an acute event, atorvastatin 80 mg reduced a composite of death from any cause, MI, documented UA requiring rehospitalization, and revascularization after 30 days of randomization or stroke when compared to pravastatin 40 mg (22.4% vs. 26.3%, HR 0.84, 95% CI 0.74 to 0.95). However, it is important to note that this endpoint is very broad and includes more fragile outcomes such as unstable angina and need for revascularization. Moreover, reductions in LDL-C levels in the pravastatin group were strikingly low (baseline LDL-C 106 mg/dL and LDL-C achieved at follow-up 95 mg/dL), which surely favors the atorvastatin group, which achieved LDL-C of 62 mg/dL on follow-up (41% reduction).

The Incremental Decrease in Endpoints Through Aggressive Lipid Lowering (IDEAL) trial tested atorvastatin 80 mg vs. simvastatin 20 mg and found no difference in the primary endpoint of coronary death, non-fatal MI, or cardiac arrest with resuscitation (HR 0.89, 95% CI 0.78 to 1.01) [23]. One limitation of this trial was the smaller than expected reductions in LDL-C levels, which were around 34% with atorvastatin 80 mg and 17.7% in the simvastatin 20 mg group, possibly blunting an eventual difference in effects, but this is merely speculatory.

The Treating to New Targets (TNT) trial tested different regimens of atorvastatin (80 mg vs. 10 mg, a high- vs. a moderate-intensity regimen) in patients with stable CAD [24]. This trial found a reduction in coronary death, nonfatal non-procedural MI, and resuscitation after cardiac arrest or stroke (8.7% vs. 10.9%, NNT of 46). The primary endpoint was mainly driven by MI and stroke.

Finally, the Study of the Effectiveness of Additional Reductions in Cholesterol and Homocysteine (SEARCH) trial tested different simvastatin doses (20 mg vs. 80 mg) and found no difference in efficacy outcomes yet with a higher incidence of myopathy [25]. Thus, simvastatin 80 mg should not be used.

Based on findings from randomized clinical trials (RCTs), it appears that high-intensity regimens lead to a modest benefit in patients with CAD, even with “normal” LDL-C levels. This benefit is mainly driven by a reduction in cardiac events, although clear mortality benefits are not evident.

**Table 2 pharmaceuticals-17-00289-t002:** Secondary prevention trials. * Included both primary and secondary prevention patients. Legend: ACS: acute coronary syndrome; CA: cardiac arrest; CAD: coronary artery disease; DM: diabetes mellitus; HTN: hypertension; MI: myocardial infarction; TC: total cholesterol; UA: unstable angina.

Study	Sample Size	Characteristics of Patients	Comparison Groups	Follow-Up	LDL-C Reduction	CV Effects
4S (1994) [16]	4444	Angina or previous MI	Simvastatin 20–40 mg vs. placebo	5.4 years	35%	Reduction in death (HR 0.70, 95% CI 0.58 to 0.85, NNT 30)
CARE (1996) [17]	4159	Previous MI TC < 240 mg/dL LDL-C 115–174 mg/dL	Pravastatin 40 mg vs. placebo	5 years	28%	Reduction in coronary death or MI (10.2% vs. 13.2%, NNT 34)
LIPID (1998) [18]	9014	Previous MI or UA TC 155–271 mg/dL	Pravastatin 40 mg vs. placebo	6.1 years	25%	Reduction in coronary death (6.4% vs. 8.3%, NNT 53)
FLORIDA (2000) [20]	540	MI	Fluvastatin 80 mg vs. placebo	1 year	21%	No significant differences in major coronary event
HPS * (2002) [19]	20,536	TC > 135 mg/dL + CAD or other arterial disease or DM or >65 years male w/HTN	Simvastatin 40 mg vs. placebo	5 years	35%	Reduction in all-cause mortality (12.9% vs. 14.7%, NNT 56)
PROVE-IT (2004) [22]	4162	ACS < 10 days	Atorvastatin 80 mg vs. pravastatin 40 mg	24 months	31%	Reduction in all-cause mortality, MI, UA hospitalization, revascularization in 30 days, or stroke (HR 0.84, 95% CI 0.74 to 0.95, NNT 53)
IDEAL (2005) [23]	8888	Previous MI	Atorvastatin 80 mg vs. simvastatin 20 mg	4.8 years	20%	No significant differences in major coronary event
TNT (2005) [24]	10,001	CAD	Atorvastatin 80 mg vs. atorvastatin 10 mg	4.9 years	24%	Reduction in CV death, MI, CA, or stroke (HR 0.78, 95% CI 0.69 to 0.89, NNT 45)
SEARCH (2010) [25]	12,064	Previous MI LDL-C > 135 mg/dL (statin use) or LDL-C > 193 mg/dL (no statin)	Simvastatin 20 mg vs. simvastatin 80 mg	6.7 years	14%	No significant differences in CV events

### 3.3. Special Groups

Despite growing evidence solidifying statins as the cornerstone in LDL-C-lowering therapy, the possibility has been raised that specific groups could have clinical benefits from statins.

The Collaborative Atorvastatin Diabetes Study (CARDS) trial tested atorvastatin 10 mg vs. placebo for primary prevention in patients with diabetes and at least one additional risk factor (retinopathy, albuminuria, current smoking, or hypertension), with LDL-C < 160 mg/dL [26]. Although truncated (terminated 2 years earlier due to prespecified efficacy criteria), this trial showed a reduction in CV events (9.0% vs. 3.2%, NNT 32).

The Atorvastatin Study for Prevention of Coronary Heart Disease Endpoints in Non-Insulin-Dependent Diabetes Mellitus (ASPEN) trial randomized patients with diabetes in primary (79%) and secondary prevention (21%) to receive atorvastatin 10 mg or placebo [27]. The trial found no difference in the primary endpoint of CV death, MI, stroke, revascularization, worsening UA requiring hospitalization, or resuscitated cardiac arrest. These results can be explained by significant steering disturbances during the trial. Widespread recognition of the importance of CV prevention in diabetes patients led to a recommendation to stop the study medications and allocate all secondary prevention patients and previously primary prevention patients who now met an endpoint to receive usual care. This resulted in a very low completion rate, with only 67% of the intervention group and 58% of the placebo group receiving the study medication at the end of the double-blind follow-up. This reduction in the number of participants reduces the power to detect differences and hinders the evaluation of the results; caution should be taken when using these findings.

Sub-analyses of the HPS and ASCOT-LLA trials with diabetic patients showed that statin reduces CV events [28,29].

Three clinical trials have evaluated the cardiovascular effects of statins in patients with chronic kidney disease [30,31,32]. None of them showed CV benefits of statins in this patient population.

On the other hand, the Study of Heart and Renal Protection (SHARP) showed that ezetimibe/simvastatin, compared to placebo, reduces LDL-C and atherosclerotic and major vascular events in patients with CAD but no overt CAD (11.3% vs. 13.4%, HR 0.83, 95% CI 0.74 to 0.94) [33].

Therefore, CKD patients who are not on dialysis might benefit from statins, with a modest impact, while CKD patients on dialysis do not appear to derive benefits from this therapy.

Patients with HF have often been excluded from statin trials. However, two clinical trials have evaluated the effects of rosuvastatin in patients with chronic symptomatic HF [34,35]. Both trials did not show a reduction in CV events in this population.

An elderly population was exclusively studied in the Prospective Study of Pravastatin in the Elderly at Risk (PROSPER) trial [36]. This study showed a reduction in the composite primary outcome of coronary death, MI, and stroke (14.1% vs. 16.2%, HR 0.85, 95% CI 0.74 to 0.97, NNT 48) in the statin group.

People with HIV infection, a group with an increased CV risk, were also analyzed in a recent phase 3 trial. In the Randomized Trial to Prevent Vascular Events in HIV (REPRIEVE) study, 7769 participants with HIV infection were randomized to daily pitavastatin at a dose of 4 mg or placebo. After a median follow-up of 5.1 years, the study was interrupted due to efficacy. The rate of MACEs was 4.81 and 7.32 per 1000 person years in the pitavastatin and placebo groups, respectively (HR 0.65, 95% CI 0.48 to 0.90, *p* = 0.002). Muscle-related symptoms and incident diabetes were more common in the pitavastatin group [37].

Finally, a meta-analysis of data from 170,000 patients evaluated statin vs. placebo and different statin regimens and reported a 10% reduction in all-cause mortality per 38 mg/dL LDL-C reduction (HR 0.90, 95% CI 0.87 to 0.93) [38]. In an unweighted analysis of the 21 placebo-controlled trials included, any major vascular event occurred in 3.6% in the placebo groups vs. 2.8% in the statin groups, translating into a 0.8% ARR and a 22% RRR. Regarding higher- vs. lower-intensity regimens, higher-intensity regimens led to reductions in major vascular events (RRR 15%, 95% CI 11 to 18), especially when weighted for LDL-C reductions (RRR per 1 mmol/L reduction in LDL-C), suggesting that greater reductions in LDL-C accompany greater reductions in MACEs.

The evidence presented so far establishes the relationship between intervention and effect, and statins reduce CV events. The magnitude of benefits should, however, always permeate clinical reasoning, and NNTs ranging from 30 to much higher numbers have been found. The higher the baseline risk, the greater benefit that should be expected from statins. There is a logical chain binding CV risk factors, CV disease, and death. Treating one will probably affect the other but with progressively smaller magnitude. On the other hand, benefits in the discussed trials above seem to increase over time, which is expected since risk factors may be lifelong cumulative. Trials tend to follow patients over the course of a few years, and potential long-term benefits should be taken into account. The concepts proven with the studies above should be used as tools to individualize decision making for each particular patient.

**Table 3 pharmaceuticals-17-00289-t003:** Statins in special groups. Legend: ACS: acute coronary syndrome; CHD: coronary heart disease; CKD: chronic kidney disease; CV: cardiovascular; DM: diabetes mellitus; LVEF: left ventricular ejection fraction; HIV: human immunodeficiency virus; MACEs: major cardiovascular events; MI: myocardial infarction; NYHA: New York Heart Association; TGs: triglycerides.

Study	Sample Size	Characteristics of Patients	Comparison Groups	Follow-Up	LDL-C Reduction	CV Effects
CARDS (2004) [26]	2838	DM (40–75 years) + LDL-C < 160 mg/dL + TGs < 600 mg/dL + additional risk factor	Atorvastatin 10 mg vs. placebo	3.9 years	40%	Reduction in ACS, revascularization, or stroke (HR 0.63, 95% CI 0.48 to 0.83, NNT 31)
ASPEN (2006) [27]	2410	Diabetes (40–75 years) + LDL < 160 mg/dL or < 140 mg/dL (prior MI or revascularization)	Atorvastatin 10 mg vs. placebo	4 years	29%	No significant differences in CV events
ALERT (2003) [30]	2102	Renal or combined renal and pancreas transplants > 6 months	Fluvastatin 40 mg vs. placebo	5.1 years	25%	No significant differences in CV events
4D (2005) [31]	1255	Diabetes + CKD on dialysis	Atorvastatin 20 mg vs. placebo	4 years	42%	No significant differences in CV events
AURORA (2009) [32]	2773	CKD on dialysis	Rosuvastatin 10 mg vs. placebo	3.8 years	43%	No significant differences in CV events
SHARP (2011) [33]	9270	CKD	Simvastatin 20 mg + ezetimibe 10 mg vs. placebo	4.9 years	31%	Reduction in coronary death, MI, stroke, or revascularization (HR 0.83, 95% CI 0.74 to 0.94, NNT 53)
CORONA (2007) [34]	5011	LVEF < 40% + NYHA II–IV	Rosuvastatin 10 mg vs. placebo	2.7 years	45%	No significant differences in CV events
GISSI-HF (2008) [35]	6975	Heart failure NYHA II–IV	Rosuvastatin 10 mg vs. placebo	3.9 years	16%	No significant differences in CV events
PROSPER (2002) [36]	5804	Elderly (70–82 years) + high CV risk	Pravastatin 40 mg vs. placebo	3.2 years	34%	Reduction in coronary death, MI, or stroke (HR 0.85, 95% CI 0.74 to 0.97, NNT 48)
REPRIEVE (2023) [37]	7769	HIV	Pitavastatin 4 mg vs. placebo	5.1 years	30%	Reduction in MACEs (HR 0.65, 95% CI 0.48 to 0.90)

## 4. PCSK9 Inhibitors

PCSK9 inhibitors constitute a relatively new class of drugs designed to lower LDL-C levels, and they have been utilized in conjunction with statins and ezetimibe for patients at very high CV risk, particularly those engaged in secondary prevention and those with familial hypercholesterolemia [FH] and additional risk factors, who have not achieved prespecified target LDL-C levels [7].

PCSK9 is a protein that binds to LDLRs in hepatocytes, resulting in the degradation of the receptor. This, in turn, leads to reduced removal of LDL-C from circulation. PCSK9 inhibitors bind to this protein, thereby increasing LDLR levels and facilitating the lowering of LDL-C [39]. As their effectiveness depends on the presence of LDLRs, patients with receptor-deficient homozygous familial hypercholesterolemia (HoFH) do not respond well to this drug class [7]. These drugs are injectable monoclonal antibodies and typically exhibit a long half-life, enabling periodic administration. The available PCSK9 inhibitors, alirocumab and evolocumab, can be administered every two weeks or even monthly.

These drugs significantly decrease LDL-C levels, typically around 60%, whether used alone or in combination with other drug classes. Similar to statins, PCSK9 inhibitors may also decrease triglycerides and induce a slight increase in HDL-C levels [7].

Additionally, an intriguing response to PCSK9 inhibitors is the reduction of Lp(a), a result not attained by most drug classes. Both alirocumab and evolocumab achieved approximately 30% reductions in Lp(a) levels [40].

The Further Cardiovascular Outcomes Research with PCSK9 Inhibition in Subjects with Elevated Risk (FOURIER) trial found a 1.5% ARR in CV death, MI, stroke, hospitalization for UA, or coronary revascularization with evolocumab (9.8% vs. 11.3%, HR 0.85, 95% CI 0.79 to 0.92, NNT 67) [41] (Table 4). However, it is important to interpret these findings in clinical practice with caution—notably, the reduction in risk itself is marginal (1.5%), considering it is a composite outcome.

The Evaluation of Cardiovascular Outcomes After an Acute Coronary Syndrome During Treatment with Alirocumab (ODYSSEY OUTCOMES) trial found a reduction in the composite outcome of coronary death, MI, stroke, and hospitalization for UA with alirocumab (9.5% vs. 11.1%, HR 0.85, 95% CI 0.78 to 0.93, NNT 63) [42] (Table 4). The ODYSSEY OUTCOMES trial also reported a reduction in the secondary outcome of death from any cause (3.5% vs. 4.1%) but without a significant reduction in CV death, which increases the odds that this finding was a result of chance. Like the FOURIER trial, the ODYSSEY OUTCOMES trial also presented a low effect estimate with PCSK9 inhibitors.

In terms of the safety of PCSK9 inhibitors, both trials yielded similar findings, and certain observations need to be highlighted. The ODYSSEY OUTCOMES trial reported a high rate (~77%) of adverse events, with no significant difference between the treatment and placebo groups. About 24% of these events were considered serious, and only ~1.5% were believed to be related to the study agent, leading to discontinuation of the treatment. Injection site reactions were more common in the evolocumab group (2.1% vs. 1.6%), and they were typically mild, resulting in discontinuation in only 0.1% of patients in each group.

Both trials aimed to achieve the same goal of further reducing LDL-C, resulting in marginal reductions in events. Clearly, it is not anticipated that new treatments will lead to substantial absolute risk reductions when used in conjunction with statins and ezetimibe, given the current context of residual risk reduction. Nevertheless, it is crucial to maintain high standards when assessing costs and potential measured and unmeasured adverse outcomes associated with prescribing a treatment that holds marginal clinical relevance. The anticipated effects of the treatment should be discussed with patients to facilitate shared decision making, integrating the treatment’s impact size on outcomes into the discussion.

The short follow-up in these trials testing add-on therapies in patients already intensively treated could also contribute to those unimpressive results. Indeed, an open-label extension study of FOURIER called FOURIER-OLE shed some light on this matter. This extension enrolled a total of 6635 patients from the FOURIER trial to receive evolocumab for an extension period, with 3355 originally randomized to evolocumab and 3280 originally randomized to placebo. The median follow-up in the FOURIER-OLE was 5 years, with the maximum exposure to evolocumab, considering FOURIER and FOURIER-OLE, of 8.4 years. At 12 weeks, the median LDL-C was 30 mg/dL. The incidence of serious adverse events, muscle events, new-onset diabetes, hemorrhagic stroke, and neurocognitive events did not exceed the incidence observed in the placebo group of the original study and did not increase over time. The risk of the composite CV outcome was 15% lower in patients originally randomized to evolocumab compared to patients originally randomized to placebo (OR 0.85, 95% CI 0.75 to 0.96, *p* = 0.008), with a 23% lower risk of CV death (HR 0.77, 95% CI 0.60 to 0.99, *p* = 0.04), suggesting that long-term LDL-C reduction additionally decreases the risk of CV events [43].

### 4.1. Tafolecimab

Tafolecimab is a fully human IgG2 PCSK9 monoclonal antibody. Phase 1 studies demonstrated that tafolecimab reduces LDL-C by more than 70% [44]. The phase 3 Clinical Research of Developing PCSK9 Inhibitor as Cholesterol-Lowering Therapy in Chinese Patients with Dyslipidemia-1 (CREDIT-1) study showed that tafolecimab was safe and exhibited superior lipid-lowering efficacy compared to the placebo in non-FH patients [45]. The Phase 3 Clinical Research of Developing PCSK9 Inhibitor as Cholesterol-lowering Therapy in Chinese Patients with Dyslipidemia-2 (CREDIT-2) study demonstrated that tafolecimab led to significant and persistent reductions in LDL-C levels and showed a favorable safety profile in Chinese patients with heterozygous familial hypercholesterolemia (HeFH) [46].

### 4.2. Lerodalcibep

Lerodalcibep is a small binding protein that inhibits interaction between PCSK9 and LDL receptors, offering an alternative to monoclonal antibodies. Phase 2 studies demonstrated that Lerodalcibep (LIB003) significantly reduced LDL-C [47,48]. The Long-term Efficacy and Safety of Lerodalcibep in Heterozygous Familial Hypercholesterolemia (LIBerate-HeFH) trial randomized 319 patients with HeFH to monthly subcutaneous 1.2 mL injections of lerodalcibep 300 mg for 24 weeks and 159 patients with HeFH to placebo [49]. Among patients who received monthly lerodalcibep injection within the steady window period (80% of the overall cohort), researchers reported a 24-week placebo-adjusted reduction in LDL-C of 63.6% (*p* < 0.0001) and a mean placebo-adjusted reduction during the average of weeks 22 and 24 of 70.2% (*p* < 0.0001) [50].

## 5. Ezetimibe

Ezetimibe binds to the Niemann–Pick C1-like 1 (NPC1L1) protein and selectively inhibits cholesterol absorption [51]. The reduction in cholesterol absorption leads to a decrease in the delivery of cholesterol to the liver, an increase in cholesterol clearance from the blood, and a decrease in hepatic cholesterol stores.

Dujovne et al. reported the efficacy of ezetimibe in 892 patients with primary hypercholesterolemia observed in 12 weeks of treatment. They found a 17% average reduction in LDL-C compared to the placebo group, as well as decreases in plasma levels of apoB, triglycerides, and a slight increase in HDL-C [52].

Table 5 summarizes the efficacy results from the main studies involving ezetimibe.

The Ezetimibe Add-On to Statin for Effectiveness (EASE) trial showed that the addition of ezetimibe to the statin resulted in an additional reduction of up to 23% in LDL-C. Seventy percent of these patients successfully achieved the objectives recommended by the National Cholesterol Education Program (NCEP) Adult Treatment Panel III (ATP III), compared to only 17% of those on standard therapy with statin alone [53].

In 2008, the Ezetimibe and Simvastatin in Hypercholesterolemia Enhances Atherosclerosis Regression (ENHANCE) trial raised doubts regarding the efficacy of ezetimibe. The trial showed that ezetimibe in combination with simvastatin did not significantly change the mean carotid artery intima-media thickness in patients with FH [54]. However, there was a significant decrease in LDL-C, triglyceride, and hsCRP levels. Ezetimibe treatment was not associated with a significant increase in adverse events compared to the placebo.

Subsequently, The Improved Reduction of Outcomes: Vytorin Efficacy International Trial (IMPROVE-IT) showed that the combination of ezetimibe 10 mg and simvastatin 40 mg reduced the incidence of CV events compared to simvastatin 40 mg, in patients with high-risk acute coronary syndrome (ACS) and low LDL-C (<125 mg/dL) [55]. However, despite the study having a NNT of 50, the RRR was only 6%. Furthermore, the confidence interval of this estimate, ranging from 1% to 11%, indicates the uncertainty of their estimates. Therefore, the IMPROVE-IT trial suggests that ezetimibe, when used in conjunction with statins in patients with reasonably low cholesterol levels (mean LDL-C = 90 mg/dL), yields a beneficial effect of minimal magnitude, yet it is proportionate to its modest impact on LDL-C reduction.

The Heart Institute of Japan Proper Level of Lipid Lowering with Pitavastatin and Ezetimibe in Acute Coronary Syndrome (HIJ-PROPER) Trial, on the other hand, showed that the combination of pitavastatin + ezetimibe vs. pitavastatin alone was not associated with a reduction in CV events (all-cause death, MI, stroke, UA or ischemia-guided coronary revascularization) in patients with non-ST elevation ACS (HR 0.89, 95% CI 0.76–1.04) [56].

The Ezetimibe Lipid-Lowering Trial on the Prevention of Atherosclerotic Cardiovascular Disease in 75 or Older (EWTOPIA 75) trial demonstrated that, compared with dietary counseling alone, the use of additional ezetimibe for primary prevention among elderly Japanese patients with LDL-C ≥ 140 mg/dL and ≥1 high-risk feature reduced CV events, primarily cardiac events, with no difference in all-cause mortality. None of these patients were on statin therapy [57].

Finally, the Randomized Comparison of Efficacy and Safety of Lipid Lowering with Statin Monotherapy Versus Statin/Ezetimibe Combination for High-risk Cardiovascular Disease (RACING) trial showed that among patients with ASCVD, moderate-intensity statin with ezetimibe combination therapy was non-inferior to high-intensity statin monotherapy for the 3-year composite outcomes (CV death, major CV events, or stroke) [58].

## 6. Bempedoic Acid

Bempedoic acid is an ATP citrate lyase (ACL) inhibitor that targets cholesterol synthesis upstream of 3-hydroxy-3-methylglutaryl coenzyme A reductase, the enzyme inhibited by statins [59]. ACL decreases the conversion of mitochondrial-derived citrate to cytosolic ATP citrate lyase, thus reducing substrate availability for cholesterol and fatty acid synthesis. The end result is a decrease in LDL-C synthesis and upregulating hepatic LDL receptor expression.

The Cholesterol Lowering via Bempedoic Acid [ECT1002], an ACL-Inhibiting Regimen (CLEAR OUTCOMES) trial showed that bempedoic acid was associated with a lower risk of major adverse CV events (CV death, MI, stroke, or coronary revascularization) among statin-intolerant patients (bempedoic acid 11.7% vs. placebo 13.3%; HR 0.87, 95% CI 0.79 to 0.96; NNT 62.5) [60].

In the context of high-CV-risk patients with elevated LDL-C, even with maximally tolerated statin or in combination with other therapies, a fixed-dose combination of bempedoic acid and ezetimibe significantly reduced LDL-C compared to placebo [61].

Furthermore, in a recent meta-analysis comprising 10 RCTs and 18,200 patients, bempedoic acid was associated with a lower risk of CV events (CV death, MI, or stroke) [OR 0.84; 95% CI 0.76 to 0.96; *p* < 0.001; I2 = 0%) [62].

Interestingly, in an analysis by Ridker et al., bempedoic acid demonstrated a similar relative efficacy compared to placebo across baseline hsCRP and LDL-C categories [56]. Additionally, the authors showed that inflammation assessed by hsCRP predicted risk for future CV events and death more strongly than LDL-C [63]. However, this analysis should not be interpreted as diminishing the critical role of lipid lowering beyond statins for patients with persistent or refractory hypercholesterolemia. Instead, it suggests that targeting LDL-C alone may not completely mitigate atherosclerotic risk, and anti-inflammatory pathways may provide incremental CV benefits.

## 7. Lp(a)-Targeted Therapies

Lipoprotein(a) [Lp(a)] was first described by Berg in 1963 as an antigenically distinct form of beta-lipoprotein, composed of an LDL-like particle to which apoB-100 is covalently linked by a single disulfide bond to apo(a), the pathognomonic component of Lp(a) [64]. There is controversy regarding the use of Lp(a) in the prevention of atherosclerotic cardiovascular disease (ASCVD); more than 90% of circulating Lp(a) levels are genetically determined, not greatly influenced by diet or behavioral measures. As a result, their levels do not fluctuate significantly throughout life, making it challenging to devise potential therapies for reducing these levels [65]. Several guidelines currently recommend measuring Lp(a) only once in patients with early ASCVD, familial hypercholesterolemia (FH), a family history of early ASCVD, and/or elevated Lp(a), recurrent cardiovascular (CV) events despite statin use and for the reclassification of individuals with borderline risk (that is, 10-year risk of a CV event below 15%) [66,67,68].

It was believed that the use of statins would not influence the fluctuation of Lp(a) levels, as LDLRs, which are more expressed due to the inhibition of HMG-CoA reductase and consequent reduction in tissue LDL-C, do not appear to play an important role in the clearance of Lp(a). However, in a study involving 3,896 patients, an average increase in Lp(a) of 11% was observed when measured before and after the initiation of statins of different strengths [69]. The mechanisms for this observed increase are not clear, but it is conjectured that it may be one of the reasons why some patients using maximum doses of statins are non-responders because the majority of their cholesterol is in Lp(a) instead of the LDL-C particles [70]. These results did not occur in the Justification for the Use of Statins in Prevention: An Intervention Trial Evaluating Rosuvastatin (JUPITER) trial [14], whose Lp(a) value was also measured before and after the use of statins (rosuvastatin), and the average variation was equal to zero. Although the highest baseline Lp(a) values—above 50 nmol/L—were associated with a 64% increase in the risk of MI, stroke, hospitalization for UA, arterial revascularization, and CV death compared to the lowest values (≤10 nmol/L), the group that used rosuvastatin had a lower incidence of CV events regardless of Lp(a) levels.

Specific medications to reduce Lp(a) are not yet available, although niacin, PCSK9 inhibitors, mipomersen, and estrogen have effects on reducing its levels. In the Atherothrombosis Intervention in Metabolic Syndrome with Low HDL/High Triglycerides: Impact on Global Health Outcomes (AIM-HIGH) trial, the use of niacin reduced mean Lp(a) compared to baseline by 19% (however, in absolute levels, there was a modest reduction from 13.5 mg/dL to 11 mg/dL) [71]. There were no clinical benefits in the subgroup of patients with the greatest reduction (fourth quartile with Lp(a) > 50 mg/dL) despite a 39% reduction when compared to baseline levels. Mipomersen and PCSK9 inhibitors reduce Lp(a) between 20% and 30%; however, since they do not exclusively reduce Lp(a), it is not possible to form a conclusion about the clinical benefits of Lp(a) reduction. In a post hoc analysis of the Heart and Estrogen/Progestin Replacement Study (HERS) trial, patients who received estrogen and progestin showed an average reduction of 5.8 mg/dL in Lp(a) vs. 0.34 mg/dL in the placebo group after one year of follow-up [72]. Patients who had a greater reduction in Lp(a) (>−8.8 mg/dL) had 38% fewer events (MI and CV death) than patients with a smaller reduction in Lp(a). However, due to the increased incidence of thromboembolic events in this study, hormonal therapy in menopausal women is not a common alternative.

Mendelian randomization studies estimate that each 60 to 100 mg/dL reduction in Lp(a) could translate into a 20% reduction in the rate of cardiovascular events, corresponding to a reduction in LDL-C of around 40 mg/dL with the use of statins. The potential benefits may be even greater with higher levels of Lp(a) and more significant reductions in its levels. Therefore, therapies specifically focused on intensive Lp(a) reduction have been developed and are currently being tested in high-risk individuals with elevated Lp(a) concentrations. Pelacarsen and olpasiran are both therapies directed against apo(a) mRNA and are able to reduce Lp(a) levels by approximately 80% and 100%, respectively. Phase 3 outcomes studies are ongoing [73,74].

## 8. Bile Acid-Binding Resins

Currently, three bile acid sequestrants (BASs) are available: (1) cholestyramine (developed in the 1950s), (2) colestipol (developed in the 1970s), and (3) colesevelam (approved in 2000).

Bile acid-binding resins are highly positively charged molecules that bind to negatively charged bile acids in the intestine, preventing their reabsorption in the terminal ileum and consequently increasing fecal excretion of bile acids. The reduction of the body’s bile acid pool stimulates the conversion of cholesterol to bile acids in the liver. Increased bile acid synthesis reduces hepatic cholesterol levels, leading to the upregulation of enzymes involved in the uptake of cholesterol from the bloodstream to the liver through a greater expression of LDLRs [75].

The first large double-blind study with an adequate sample size and quality to evaluate the effect of LDL-C-lowering drugs on the occurrence of CV events was the Lipid Research Clinics Coronary Primary Prevention Trial (LRC-CPPT) [76]. The trial included 3806 asymptomatic men between 35 and 59 years of age diagnosed with primary hypercholesterolemia (defined as LDL-C ≥ 190 mg/dL), who had never experienced a heart attack, angina, or heart failure. The participants were followed for an average of 7.4 years, with a mean baseline LDL-C of 216 mg/dL. The trial showed that cholestyramine reduced LDL-C by 20.3%, representing a 12.6% greater reduction than that observed in the placebo group. There was a 19% lower occurrence of primary events (CV death or MI) in the cholestyramine group.

Hypothetically, if there was a lack of adherence to treatment in the cholestyramine group for five years between the measurement of LDL-C in the first year and the measurement in the seventh year, with a return to treatment one year earlier, a faithful analysis of LDL-C behavior during the five years of poor adherence would be impossible. Instead, only the record of the shortest period of adherence would be available. While this is an extreme example, it serves to illustrate the caution required when interpreting studies that assess associations based on widely spaced LDL-C measurements.

Colestipol can reduce LDL-C levels by 15 to 30% [77], but there are no published clinical trials that have evaluated the benefit of colestipol in reducing CV events. Currently, its use is restricted to some cases of patients with primary hypercholesterolemia. Colesevelam monotherapy was studied in two clinical trials of patients with moderate primary hypercholesterolemia. A maximum LDL-C reduction of 19% compared to baseline was achieved after 2 weeks of treatment [78,79].

Meta-analysis using Mendelian randomization evaluated the effect of cholestyramine and colesevelam by quantifying the effect on CV risk reduction of rs4299376 (ABCG5/ABCG8), which affects the intestinal absorption pathway of BAS target cholesterol [80]. Nineteen RCTs with a total of 7021 study participants were included. Cholestyramine was associated with a reduction in LDL-C of 23.5 mg/dL and a trend towards a reduced risk of CAD (OR 0.81, 95% CI 0.70 to 1.02; *p* = 0.07), while colesevelam was associated with a reduction in LDL-C of 22.7 mg/dL (95% CI −28.3 to −17.2). There is no description of baseline LDL-C in these studies. Based on the findings that rs4299376 was associated with a 2.75 mg/dL decrease in LDL-C and a 5% decrease in the risk of CAD outcomes, cholestyramine was associated with an OR for CAD of 0.63 (95% CI 0.52 to 0.77; *p* = 6.3 × 10^−6^) and colesevelam with an OR of 0.64 (95% CI 0.52 to 0.79, *p* = 4.3 × 10^−5^).

Due to the modest reduction in LDL-C and undesirable gastrointestinal effects, these medications remain in the background in the treatment of dyslipidemia. Additionally, they are used for the treatment of type 2 diabetes mellitus, as they have been shown to lower glycosylated hemoglobin (A1C) by an average of 0.5% in patients [80].

## 9. Nicotinic Acid

Niacin or nicotinic acid is a soluble vitamin with lipid-lowering properties. Niacin appears to reduce the mobilization of free fatty acids from adipocytes, acting on specific receptors and thereby reducing the formation of lipoproteins rich in triglycerides (TGs) in the liver. There are two forms of niacin: one with rapid absorption (crystalline), more commonly associated with flushing, and another with extended release, with better tolerability.

Niacin is capable of increasing HDL-C levels and reducing concentrations of apoB-containing lipoproteins [VLDL, IDL, LDL and Lp(a)]. The medication was evaluated in several clinical trials, as shown in Table 6.

A meta-analysis of 17 studies with niacin that provided data on cardiovascular disease (CVD) outcomes showed that niacin might have some utility in lipid control for secondary prevention as monotherapy, although the evidence is from older studies in a population potentially not representative of current patients [89].

However, Ronsein et al. recently showed that the addition of niacin to statin therapy resulted in elevated levels of multiple HDL proteins linked to increased atherosclerotic risk, which might have compromised the cardioprotective effects associated with higher HDL-C levels and lower levels of LDL-C and TGs [90].

## 10. Fibric Acid Derivatives

Fibrates have multiple pharmacological actions, mainly as a synthetic ligand for the peroxisome proliferator-activated receptors (PPARs), especially PPARα. Clinically, fibrates reduce plasma triglycerides (TGs) or TG-rich lipoproteins (TRLs) and increase HDL-C levels.

Fibrates were evaluated in several clinical trials as shown in Table 7.

A meta-analysis of six trials, including 16,135 individuals, evaluated the clinical benefits of fibrates for the primary prevention of CV disease. It showed that fibrates lower the risk of CV and coronary events in primary prevention, although the absolute treatment effects in the primary prevention setting were modest (ARR < 1%) [99].

In the context of secondary prevention, a meta-analysis of 13 RCTs involving 16,112 patients provided evidence of a protective effect of fibrates compared with placebo concerning the primary composite outcome of stroke, MI, and CV death (HR 0.88, 95% CI 0.83 to 0.94) [100]. Nevertheless, recent studies, particularly FIELD and ACCORD, did not offer further scientific support for the use of fibrates in cardiovascular prevention. On the other hand, both studies did not specifically include baseline TG levels as an inclusion criterion. A subsequent subgroup analysis showed a benefit in reducing CV risk in those with hypertriglyceridemia compared to individuals with lower TG values. The Pemafibrate to Reduce Cardiovascular Outcomes by Reducing Triglycerides In Patients with Diabetes (PROMINENT) was conducted to evaluate the role of fibrate additive therapy in reducing the residual risk of ASCVD in patients with high TGs [98]. Pemafibrate is a synthetic ligand to PPAR-alpha, with potency greater than 2500 times that of fenofibrate and with greater selectivity for PPAR-alpha in relation to PPAR-gamma. In 2022, the study was interrupted due to futility. These results suggest that the fibrates do not provide a sufficient level of evidence for their use in CV prevention in patients with mild to moderate hypertriglyceridemia (TGs between 150 and 500 mg/dL).

## 11. Omega-3 Fatty Acids

Omega-3 fatty acids (OM3FAs) are unsaturated fatty acids with at least one double bond located between the third and fourth omega end carbon. Currently, the three most clinically relevant omega-3 polyunsaturated fatty acids (PUFAs) are α-linolenic acid (ALA), eicosapentaenoic acid (EPA), and docosahexaenoic acid (DHA). OM3FAs suppress lipogenic gene expression by decreasing the expression of sterol regulatory element-binding protein 1c, inhibiting phosphatidic acid phosphatase, and acyl-CoA:1,2-diacylglycerol acyltransferase (NGAT). Sterol regulatory element-binding proteins (SREBPs) are membrane-bound enzymes that, when cleaved, travel to the nucleus to transcribe enzymes involved in cholesterol, LDL, and fatty acid synthesis. When a diet is high in omega-3 fatty acids, the SREBPs (particularly 1c) are not activated because of negative feedback inhibition, lowering SREBP synthesis and the cholesterol synthesizing enzymes that it regulates: FPP synthase (farnesyl diphosphate synthase) and HMG-CoA reductase (3-hydroxy-3-methylglutaryl-CoA reductase) [101].

Table 8 summarizes the efficacy results from the main studies involving omega-3 fatty acids.

Two initial studies, the Japan EPA Lipid Intervention Study (JELIS) [102] and OMEGA [103], showed controversial results on the benefits of omega-3 in patients without and with established CV disease, respectively.

The Vitamin D and Omega-3 Trial (VITAL) trial showed that n-3 fatty acids did not result in a lower incidence of major CV events than placebo after a median follow-up of 5.3 years (HR 0.92; 95% CI 0.80 to 1.06; *p* = 0.24) [104].

The Multinational Reduction of Cardiovascular Events with Icosapent Ethyl—Intervention Trial of Icosapent Ethyl (REDUCE-IT) trial showed a 25% RRR in CV events (17.2% vs. 22.0%, HR 0.75, 95% CI 0.68 to 0.83, NNT 21) [105].

Recently, a post hoc exploratory analysis of patients from the REDUCE-IT study assessed whether the magnitude of the reduction in CV outcomes observed was not caused by possible harm from mineral oil, used as placebo in the comparator group [106]. In this group, levels of biomarkers associated with atherosclerosis increased over time. In other words, in the placebo group, at 12 months of follow-up, significant increases in homocysteine, Lp(a), interleukin-6, lipoprotein-associated phospholipase A2 (Lp-PLA2), C-reactive protein, and beta-1 were observed. These changes remained similar at the end of 24 months. However, it should be noted that the REDUCE-IT study was designed for hard outcomes and that post hoc analysis with biomarkers (exploratory outcomes) may raise hypotheses but do not prove that mineral oil made a difference in the study. Finally, these data may call the results of the REDUCE-IT study into question, and caution is needed until definitive evidence is obtained.

The Residual Risk with Epanova in High-Cardiovascular-Risk Patients with Hypertriglyceridemia (STRENGTH) trial included 13,078 individuals with established ASCVD and showed that there was no significant difference in the primary endpoint (CV death, MI, stroke, coronary revascularization, or UA requiring hospitalization) between the treatment (4g daily of omega-3 carboxylic acid [EPA plus DHA]) and placebo arms (12.0% vs. 12.2%; HR 0.99, 95% CI 0.90 to 1.09, *p* = 0.84) [107].

The Omega-3 Fatty Acids in Elderly with Myocardial Infarction (OMEMI) trial assessed the efficacy of 1.8 g daily of n-3 PUFA (930 mg EPA and 660 mg DHA) compared with placebo among older adults (mean age 75 years) after acute MI. The trial also showed that there was no significant difference between the n-3 FA treatment group compared with the placebo control (21.4% vs. 20.0%; HR 1.08, 95% CI 0.82 to 1.41, *p* = 0.06) [108].

The reasons for the contrasting results in CV outcomes of REDUCE-IT compared with STRENGTH and OMEMI are likely multifactorial, including patient selection, differences in endpoints, dosing, and choice of placebo.

A meta-analysis, including 149,051 participants, showed that OM3FAs were associated with reduced CV events (CV death, MI, CAD, and coronary revascularization) [109].

**Table 8 pharmaceuticals-17-00289-t008:** Characteristics of the main published trials involving omega-3 fatty acids. Legend: ACS: acute coronary syndrome; CA: carboxylic acid; CV: cardiovascular; EPA: eicosapentaenoic acid; MI: myocardial infarction; n-3 PUFAs: marine n-3 polyunsaturated fatty acids; NA: not evaluated; SCD: sudden cardiac death; TC: total cholesterol.

Study	Sample Size	Characteristics of Patients	Comparison Groups	Follow-Up	Lipids Effect	CV Effects
JELIS (2007) [102]	18,645	TC > 250 mg/dL	EPA (1.8 g/d) + statin vs. only statin	4.6 years	Reduction LDL-C Reduction TC (no differences between groups)	Reduction in CV events (HR 0.81, 95% CI 0.69 to 0.95, NNT 143)
OMEGA (2010) [103]	3818	ACS	Omega-3 (1 g/d) vs. placebo	1 year	NE	No significant differences in SCD
VITAL (2019) [104]	25,871	No known CV disease	Omega-3 (1 g/d) vs. placebo	5.3 years	NE	No significant differences in CV events
REDUCE-IT (2019) [105]	8179	Age >45 years + CV disease or age > 50 years + diabetes and ≥1 risk factor + TGs 150–499 mg/dL + LDL-C 41–100 mg/dL	Icosapent ethyl (4 g/d) vs. placebo	4.9 years	Reduction LDL-C 3.1% Reduction TGs 18.3%	Reduction in CV death, MI, stroke, coronary revascularization, or UA (HR 0.75, 95% CI 0.68 to 0.83, NNT 21)
STRENGTH (2020) [107]	13,078	High CV risk	Omega-3 CA (4 g/d) vs. placebo	42 months	NE	No significant differences in CV events
OMEMI (2021) [108]	1027	Aged 70–82 years + recent MI (2–8 weeks)	n-3 PUFA (1.8 g/d) vs. placebo	2 years	NE	No significant differences in CV events

## 12. Cholesteryl Ester Transfer Protein (CETP) Inhibitors

Cholesteryl ester transfer protein (CETP) is a plasma glycoprotein that mediates the transfer of cholesteryl esters from HDL to the apoB-containing lipoproteins, with a balanced transfer of TGs [110]. The inhibition of CETP results in an accumulation of cholesterol esters in HDL, thus resulting in increased HDL-C. Table 9 summarizes the efficacy results from the main studies involving CETP inhibitors.

The first large study that evaluated the CV effects of CETP inhibitors was the Investigation of Lipid Level Management to Understand its Impact in Atherosclerotic Events (ILLUMINATE) [111]. This study showed an increase in CV events with the medication, probably due to increased blood pressure. Subsequently, other medications in the same class had negative results in large phase 3 trials. The dal-OUTCOMES study did not show a reduction in CV events, despite improving the lipid profile, with dalcetrapib [112]. Similar findings were found with evacetrapib in the Assessment of Clinical Effects of Cholesteryl Ester Transfer Protein Inhibition with Evacetrapib in Patients at a High Risk for Vascular Outcomes (ACCELERATE) trial [113].

On the other hand, the phase 3 Randomized Evaluation of the Effects of Anacetrapib through Lipid Modification (REVEAL) trial showed a reduction in the primary composite outcome with anacetrapib [114], although with an RRR of only 9% and a modest NNT of 100, demonstrating the beneficial effect of minimal magnitude. Despite the trial’s positive results, a few weeks after its publication, Merck announced that it would not attempt to approve the medication at the U.S. Food and Drug Administration (FDA).

Obicetrapib, although belonging to the class of CETP inhibitors, has attracted interest due to its potent LDL-C reduction and favorable safety profile. In the Randomized Study of Obicetrapib as an Adjunct to Statin Therapy (ROSE) study, treatment with obicetrapib for 8 weeks when compared with placebo in 120 individuals with dyslipidemia on statin treatment, demonstrated a reduction in LDL-C, apoB, and non-HDL-C by 51%, 30%, and 44% compared to placebo, respectively [115]. Obicetrapib has also been evaluated in combination with ezetimibe in individuals treated with statins in a phase 2 study, showing a 63% reduction in LDL-C. The Cardiovascular Outcomes Study to Evaluate the Effect of Obicetrapib in Patients with Cardiovascular Disease (PREVAIL), which aims to evaluate obicetrapib in about 9000 patients with ASCVD, is ongoing (NCT05202509) [116].

## 13. Gene Therapy

Gene therapy can be conducted by replacing a missing gene, overexpressing a specific gene, interfering with gene expression, or promoting the repair of a gene carrying a mutation. Several strategies can be employed to repair a gene, but crucial to the success of this therapy is the method of transport to the target cell and its nucleus. Viral vectors have been widely used as carriers, but this method has limitations, including hindering large-scale production and the risk of contamination by other viruses, among other concerns.

The emergence of new technologies, such as small interfering ribonucleic acid (siRNA), antisense oligonucleotides (ASOs), clustered regularly interspaced short tandem repeats (CRISPRs), and new transport methods, such as nanomaterials and lipid carriers, has significantly advanced the clinical applicability of this method of treatment.

The primary defect in 85% of FH cases is a mutation or deletion of the LDLR-encoding gene responsible for removing LDL-C via endocytosis and intracellular degradation. However, various targets have been explored to reduce LDL-C levels, including LDL receptors, PCSK9, ANGPTL3, APOC3, and Lp(a).

The first gene therapy applied to hypercholesterolemia showed a 15% reduction in serum LDL-C concentration in only three patients [117]. However, the low efficiency of genetic reconstitution (5–10% of cultured cells incorporated the LDLR cDNA) using the retroviral vector was considered the major hurdle that led to this relatively small effect.

Since then, different strategies have been applied to increase the therapeutic efficacy or reduce the immunogenicity of gene therapy, including the following mechanisms: (1) Virus vector-mediated therapy; (2) exosome-mediated therapy; (3) siRNA; (4) ASOs; (5) minicircle DNA vectors; (6) microRNAs; (7) long non-coding RNAs (lncRNAs); (8) CRISPR/Cas9 System [118].

### 13.1. siRNA

Small interfering RNAs (siRNAs) are short regulatory RNA molecules that suppress genes that are overexpressed in a given disease through post-transcriptional silencing.

Effective pharmacological use of siRNA requires “carriers” that can deliver the siRNA to its intended site of action, such as viral and non-viral vectors (polymeric or lipid nanoparticles, polyplexes, lipoplexes, liposomes, or multifunctional nanocarriers).

In the cytoplasm, a siRNA is processed from a double-stranded RNA, which comes from the endogenous transcription of DNA or from exogenous sources such as a virus. This double-stranded RNA is then cleaved by the ATP-dependent RNA endonuclease, Dicer, into fragments 21–23 nucleotides long with two nucleotide overhangs at both ends. This siRNA is then loaded into another protein, Argonaute. Argonaute has four different domains—N-terminal, PAZ, Mid, and PIWI. Its PIWI domain has RNase activity that allows Argonaute to cleave target mRNA. The Argonaute–siRNA complex then binds with a helicase and other proteins to form the RNA-induced silencing complex (RISC). In RISC, the sense strand is separated from the antisense, or guide strand, which is catalyzed by the helicase. The sense strand is degraded in the cytoplasm, and the guide strand directs the RISC to a complementary target mRNA.

The fate of the target mRNA is determined by whether the guide mRNA exhibits optimal or sub-optimal base pairing with the target mRNA. If the guide strand shows ideal base pairing with the target mRNA, Argonaute cleaves the target mRNA. Then, the RISC complex is reused to target another mRNA. By contrast, if the guide strand exhibits suboptimal base pairing with the target mRNA, Argonaute will not cleave the mRNA. Instead, it leads to translation arrest, as the RISC complex obstructs ribosome binding and translocation. This mRNA is then guided to processing bodies (P-bodies) where it is gradually degraded. In the nucleus, siRNAs can silence transposable DNA elements, preventing their unwanted and potentially dangerous random insertions into the genome.

However, there are challenges to the administration of siRNAs [119], as illustrated in Figure 2.

#### 13.1.1. Lepodisiran

Lepodisiran is a siRNA directed at the hepatic production of apolipoprotein(a). In a phase 1 study of 48 participants with elevated Lp(a) levels and without cardiovascular disease, lepodisiran reduced Lp(a) levels by 94% at 48 weeks [120].

#### 13.1.2. Inclisiran

Since the discovery of the interaction of PCSK9 with LDL-C metabolism, several studies have sought alternative therapies for LDL-C reduction by targeting PCSK9. The initial results of monoclonal antibodies (alirocumab and evolocumab) gained significant attention in the scientific world due to the substantial decrease in LDL-C levels and reduction in CV death, MI, stroke, and hospitalization for angina over an average follow-up of 2.5 years [41,42]. However, there are lingering uncertainties about potential adverse immunological effects of these medications in the long-term, given that the studies are recent. Additionally, the current dosage (biweekly application) of these medications may pose challenges for patient adherence. Due to these concerns, an alternative to monoclonal antibody treatment was sought, still involving PCSK9 as a target.

Inclisiran is a small interfering RNA (siRNA), which is small synthetic molecule that binds to PCSK9 mRNA, promoting its degradation and the consequent inhibition of PCSK9 synthesis [121]. These molecules called siRNA are about 25 nucleotides of RNA, which have recently become known as important regulators of the expression and function of the human genome. Long double-stranded RNA, like that present in some types of viruses, induces an immunological response via interferon, which, in turn, causes a generalized interruption of protein synthesis. Due to this characteristic, long double-stranded RNA cannot be used to silence specific genes. On the other hand, siRNAs can escape the response radar via interferon and, therefore, promote the effective silencing of specific genes through complementary nucleotide sequences that affect post-transcriptional mRNA degradation, thus preventing the translation process [122]. Inclisiran is composed of these long-acting synthetic siRNA molecules, which bind intracellularly to the RISC, cleaving mRNA that would specifically participate in the translation of PCSK9 and consequently its synthesis in the liver, allowing the reduction in circulating levels of PCSK9 by up to 70% in phase 1 study [123].

The characteristics of the main trials published on inclisiran are available in Table 10. The first phase 2 study was published in 2017 [124]. The ORION-1 trial included patients with ASCVD and elevated LDL-C (>70 mg/dL) despite maximum tolerated statin therapy and patients without ASCVD but with high CV risk conditions such as diabetes and FH in whom LDL-C was >100 mg/dL despite maximally tolerated statin therapy. Patients using PCSK9 inhibitors were excluded. Patients were randomized into eight treatment groups: a single dose regimen of 200, 300, or 500 mg inclisiran, or placebo; or a two-dose regimen of 100, 200, or 300 mg inclisiran, or placebo. The primary endpoint of LDL-C reduction at 180 days was reduced by 27.9–41.9% with one subcutaneous injection and by 35.5–52.6% with two injections (*p* < 0.001). At 240 days, the reductions in PCSK9 and LDL-C remained significantly lower than baseline with all the studied doses of inclisiran. Two injections of the 300 mg dose of inclisiran produced the greatest reduction in LDL-C, with 48% of patients receiving this dose achieving an LDL-C level < 50 mg/dL.

The ORION-3 trial, an open-label extension of the phase II ORION-1 trial, assessed whether LDL-C level reduction was sustained over a four-year follow-up [125]. Patients treated with inclisiran achieved an average 47.5% reduction in LDL-C from baseline (day 1 of ORION-1) to day 210 (95% Cl −50.7 to −44.3) and a time-averaged reduction in LDL-C of 44.2% over the 4 years through twice-yearly dosing.

On the other hand, the phase 3 ORION-5 trial showed that inclisiran treatment did not reduce LDL-C levels in patients with HoFH despite substantial lowering of PCSK9 levels [126].

Subsequently, the ORION-9 trial evaluated the safety and efficacy of inclisiran in lowering LDL-C among patients with HeFH [127]. There was a 50% observed LDL-C lowering at Day 510 with a 45% time-adjusted LDL-C lowering between days 90 and 540, indicating that inclisiran is superior to placebo in reducing LDL-C among patients with HeFH who are already on statins and ezetimibe. Almost all patients in both groups used statins (90%); however, 76.4% of the group that received inclisiran used high-potency statins vs. 71.2% of the placebo group, and 55.8% of the inclisiran group used ezetimibe vs. 50.0% of the placebo group. These differences in lipid-lowering treatment between the groups do not seem to have considerably affected the result, but the increase of 8.2% in LDL-C levels on day 510 when compared to baseline levels in the placebo group is noteworthy, even when receiving a high-potency statin in a large proportion and half of the population using ezetimibe.

A pooled analysis of two phase 3 trials (ORION-10 and -11 trials) evaluated the individual responses of patients on LDL-C reduction with inclisiran [128]. This analysis showed a highly consistent effect, with a safety and tolerability profile similar to placebo, on a twice-yearly dosing schedule after an initial dose and one 3 months later, across individual patients with ASCVD or risk equivalents over 17 months of treatment. At day 510, inclisiran reduced LDL-C levels by 52.3% (95% CI 48.8 to 55.7) in the ORION-10 trial and by 49.9% (95% CI 46.6 to 53.1) in the ORION-11 trial.

In a post hoc analysis of the ORION-9, -10 and -11 trials showed that inclisiran was well tolerated and effective in LDL-C reduction in both patients with and without polyvascular disease (PVD) [129]. The mean placebo-corrected LDL-C % change from baseline to day 510 was −48.9% (95% CI −55.6 to −42.2) in patients with PVD and −51.5% (95% CI −53.9 to −49.1) in patients without PVD.

Recently, a patient-level, pooled analysis of ORION-9, -10, and -11 trials showed a placebo-corrected percentage reduction in LDL-C with inclisiran (50.6% at day 90) [130]. Furthermore, inclisiran reduced composite MACEs (non-adjudicated CV death, cardiac arrest, MI, and stroke) [OR 0.74, 95% CI 0.58 to 0.94] but not fatal and non-fatal MIs [OR 0.80, 95% CI 0.50 to 1.27] or fatal and non-fatal stroke [OR 0.86, 95% CI 0.41 to 1.81] over 18 months, suggesting a potential benefit of inclisiran for MACEs.

The ongoing ORION-4 trial recruited more than 16,000 patients from 2018 to evaluate the reduction of clinical outcomes in patients at high risk of ASCVD [131].

### 13.2. ANGPTL3 Inhibitors

ANGPTL3 is a type of protein from the group of angiopoietin-like proteins, produced exclusively by the liver and which acts by inhibiting lipoprotein lipase (LPL)—the enzyme responsible for catalyzing the hydrolysis of plasma TGs and endothelial lipase, which appears to have a fundamental role in lipoprotein metabolism, cytokine expression, and cellular lipid composition [132]. Two other angiopoietin-like proteins act in a complementary way to ANGPTL3: ANGPTL4, which is produced in different types of cells and acts mainly to inhibit LPL during fasting; and ANGPTL8, produced by the liver and adipose tissue, which, unlike ANGPTL3 and ANGPTL4, does not have the property of inhibiting LPL on its own but can contribute to the increase or decrease in LPL inhibition through the complexes formed. The ANGPTL3/ANGPTL8 complex substantially increases affinity with LPL and creates a potent inhibitor of plasma TG clearance, while the ANGPTL4/ANGPTL8 complex neutralizes the inhibitory action of the ANGPTL3/ANGPTL8 complex [133,134,135].

A preclinical study of a monoclonal antibody targeting the C-terminal LPL inhibitory domain of ANGPTL3 was published in 2015, which resulted in increased LPL activity and reduced TGs, LDL-C, and HDL-C in rats and monkeys [136].

Loss-of-function genetic variants of ANGPTL3 are associated with low levels of LDL-C, TGs, and a lower risk of CAD despite the presence of low levels of HDL-C. In the DiscovEHR human genetics study, loss-of-function variants in ANGPTL3 were found in 0.33% of case patients with CAD and in 0.45% of controls (adjusted OR 0.59, 95% CI 0.41 to 0.85, *p* = 0.004) [137].

In 2020, the Evinacumab Lipid Studies in Patients with Homozygous Familial Hypercholesterolemia (ELIPSE HoFH) trial evaluated the efficacy of evinacumab, a monoclonal antibody against ANGPTL3, in reducing LDL-C in patients with a genetic or clinical diagnosis of HoFH [138]. Despite including patients without the need for genetic testing to confirm HoFH (32% of individuals had a clinical diagnosis), only 65 patients were included and randomized in a 2:1 ratio to receive an intravenous infusion of evinacumab or placebo every 4 weeks. At week 24, patients in the evinacumab group had a relative reduction from baseline in LDL-C of 47.1%, compared with an increase of 1.9% in the placebo group. Almost all trial patients (94%) were receiving a statin (a high-intensity statin in 77%). Additionally, a PCSK9 inhibitor was being administered in 77% of the patients, ezetimibe in 75%, and lomitapide in 25%; 34% of the patients underwent apheresis. A total of 63% of the patients were taking at least three lipid modifying drugs. Even so, the placebo group showed a 1.9% increase in LDL-C when compared to baseline after 6 months. These data strengthen the hypothesis that current therapy (mainly LDLR-targeting therapies) is a little effective or not effective for treating HoFH, especially those with null/null mutations.

The use of evinacumab was also evaluated in patients with refractory hypercholesterolemia, defined as an LDL-C level of 70 mg/dL or higher with clinical ASCVD or a level of 100 mg/dL or higher without clinical ASCVD and refractory to treatment with a PCSK9 inhibitor and a statin at a maximum tolerated dose, with or without ezetimibe; 72% of the sample had a diagnosis of HeFH [139]. After week 16 of use of evinacumab or placebo, both subcutaneous and intravenous administration of evinacumab resulted in a reduction of between 40 and 50% of LDL-C compared to baseline levels, vs. an increase of 8.8% in the placebo group of the subcutaneous regimen and 0.6% of the placebo group of the intravenous regimen. However, this trial was small and underpowered for clinical outcomes.

To date, all clinical trials involving evinacumab have only evaluated the drug’s efficacy and safety over a short period of time. These studies have confirmed the findings from animal models that the medication is not only capable of reducing LDL-C but also TGs and HDL-C. The implications of concomitant HDL-C reduction remain to be explained, but in patients with genetic ANGPTL3 deficiency, reduced HDL-C levels are not associated with an increased risk of CV disease [137].

### 13.3. CRISPR/Cas9

In 2012, Doudna and Charpentier developed the gene editing technique called CRISPR/Cas9 [140]. CRISPR/Cas9 is the name of a molecular biology technique capable of editing (removing, adding, exchanging) DNA sequences located in any region of the genome. This technique is based on an immunological memory system present in bacteria, used to protect them from virus invasion [141,142].

The Clustered Regularly Interspaced Short Palindromic Repeat Associated System 9 (CRISPR/Cas9) is a promising tool for clinical applications in the treatment of genetic diseases such as FH [143].

Zhao et al. utilized a recombinant adeno-associated virus (AAV) vector carrying the CRISPR/Cas9 gene editor (AAV-CRISPR/Cas9) in LDLR loss-of-function mutant mice that exhibited severe atherosclerotic phenotypes when fed a diet rich in fat in an in vivo animal study [144]. AAV-CRISPR/Cas9-mediated gene editing partially corrected the point mutation in the LDLR gene expressed in hepatocytes, restored partial expression of the LDLR protein, and significantly reduced total cholesterol, TGs, and LDL-C.

Wang et al. showed, through a LDLR-deficient mouse model (Ldlr^−/−^, Apobec1^−/−^, double knockout), that AAV carrying an LDLR transgene at vector doses as low as 3 × 10^11^ increased transgene expression and decreased LDL-C [145].

A recent study investigated whether a CRISPR base-editing medicine designed to alter a single DNA base in the PCSK9 gene is safe and effective in a nonhuman primate and rodent model [146]. The authors showed that a single intravenous infusion of the drug was well tolerated and was associated with an 83% reduction in PCSK9 protein and a 69% lower LDL-C, with durable effects up to 476 days after dosing.

### 13.4. Antisense Oligonucleotides (ASO)

ASO are analogues of single chains of nucleic acids, complementary to a specific mRNA (Watson–Crick hybridization), capable of inhibiting its expression and blocking protein synthesis. This technology is used to treat different diseases, including hypercholesterolemia.

#### 13.4.1. Mipomersen

Apolipoprotein B (apoB) is the protein component of the lipoproteins considered atherogenic: VLDL-C, IDL-C, and LDL-C. There are two forms of apoB: apoB-49, which is synthesized in enterocytes and is present in chylomicrons, and apoB-100, which is synthesized in hepatocytes and is present in VLDL and, consequently, in IDL and LDL.

A mutation in apoB that leads to a reduction in its synthesis, as in hypobetalipoproteinemia, also results in a reduction in LDL-C levels. Therefore, inhibiting apoB synthesis could be a therapeutic target for reducing LDL-C.

Mipomersen is a second-generation antisense oligonucleotide that binds to the messenger RNA-encoding apoB 100, thereby reducing its production.

A post hoc analysis of collected data from three RCTs showed that long-term mipomersen reduced CV events in patients with FH [147].

On the other hand, meta-analysis including 5 RCTs showed that mipomersen probably reduces LDL-C compared with placebo (mean difference: −24.79, 95% CI −30.15 to −19.43) but with a moderate level of certainty [148].

#### 13.4.2. Volanesorsen and Olezarsen

Volanesorsen, a second-generation ASO, binds at base position 489–508 of the ApoC-III mRNA, allowing for ribonuclease H1-mediated RNA degradation and consequently decreased ApoC-III synthesis. Its lipid-lowering action is LPL-independent [149].

The APPROACH study showed a mean decrement of TG levels of 77% in familial chylomicronemia patients [150]. The COMPASS study achieved similar results in patients who had TG levels > 500 mg/dL [151].

The BROADEN trial assessed the impact of the drug in 15 familial partial lipodystrophy patients. TG levels decreased by 69%. Insulin action increased by 50%, and hemoglobin A1c levels decreased by 0.44% [152].

Olezarsen is another ASO targeting APOC-III, but, differently from Volanesorsen, it is an N-acetyl-galactosamine (GalNAc)-conjugated ASO, presenting higher binding capacity and affinity with hepatic receptors. Therefore, lower doses may be necessary for a clinically significant effect, favoring safety. In a phase 2 study including individuals with high TGs, olezarsen promoted reduction in TGs in up to 60% relative to placebo, with mild injection-site reactions being the most common adverse event [153].

### 13.5. apoB and MTP Inhibitors

In recent years, the value of apoB (an essential structural component of LDL-C) as a marker of CV risk has increased through epidemiological studies, clinical trials, and Mendelian randomization analyses [154,155,156,157,158,159,160,161]. Based on this evidence, the European Society of Cardiology recommended apoB as a more accurate marker of CV risk when compared to LDL-C and non-HDL-C and a more precise parameter for lipid-lowering therapy [7]. MTP mediates triglyceride absorption and chylomicron secretion from the intestine and very-low-density lipoprotein (VLDL) secretion from the liver by linking lipid molecules with apolipoprotein B (apoB). The inhibition of MTP reduces the level of all apoB-containing lipoproteins, including LDL.

In 2006, the first human study of an antisense oligonucleotide targeting hepatic apoB mRNA to inhibit its synthesis was published, which was later named mipomersen. The reduction in apoB compared to baseline after 83 days of the first dose was 50% [162]. A meta-analysis observed an average reduction of 33% in apoB with the use of mipomersen, associated with reductions in LDL-C, non-HDL-C, TGs, and Lp(a) [163]. More recently, another meta-analysis observed a 26% reduction in LDL-C in patients using mipomersen [164].

Although there was no significant discontinuation in the first study (even with 72% of the population presenting injection-site reactions), when used in studies with a larger number of patients and longer treatment duration, important adverse effects were consistently associated with mipomersen, since injection-site reactions, such as flu-like symptoms and liver injury due to steatosis, which compromise its widespread use [164].

### 13.6. Inducible Degrader of LDLRs (IDOL)

IDOL (inducible degrader of LDLRs) is an E3 ubiquitin ligase (a kind of enzyme that can covalently combine with the substrates of various ubiquitin proteins and promotes the degradation of the substrate proteins) regulated by liver X receptor (LXR), which promotes the ubiquitination and degradation of LDLRs through lysosomal degradation by binding to the cytoplasmic region of LDLRs, regulating cholesterol metabolism through the LXR-IDOL-LDLR axis [165]. The activation of LXR by oxysterol ligands increases the transcription of genes whose protein products work to reduce intracellular cholesterol levels [166]. Treatment of various cell types, including macrophages and hepatocytes with LXR agonists, markedly inhibits the binding and uptake of LDL by these cells. LXR activation was shown to lead to rapid elimination of the LDLR protein from the cell surface. The importance of the LXR-IDOL-LDLR axis in vivo was provided by the observation that LXR agonists reduce LDLR protein levels in mice in a tissue-specific manner depending on the degree of IDOL induction [167].

Since the discovery of the IDOL gene in 2009 [165], population-based observational studies have shown that its genetic variants are strongly associated with plasma lipid levels [165,168]. Consequently, the search for compounds capable of silencing IDOL and consequently reducing LDLR degradation is ongoing.

## 14. Lomitapide

A microsomal triglyceride transfer protein (MTP) is a product of the MTTP gene and is essential for the mounting and secretion of apoB-containing lipoproteins. Genetic MTP deficiency is associated with abetalipoproteinemia, characterized by the virtual absence of apoB-containing lipoproteins in the circulation. Lomitapide is a selective inhibitor of MTP, responsible for transferring neutral lipids, predominantly TGs, to triglyceride-rich lipoproteins (TGRLs) intracellularly—chylomicrons in the intestine and very low-density lipoproteins (VLDLs) in hepatocytes [169,170].

MTP becomes an attractive therapeutic target because it potentially inhibits the production of TGRLs at the source, decreasing the formation of chylomicrons in the intestine and VLDLs in the liver, without affecting other targets such as LDLRs and LPL. However, this inhibition can lead to asymmetrical harmful effects, causing damage due to the reduced absorption of fats (resulting in diarrhea and gastrointestinal symptoms) and the increased formation of VLDLs in the liver (leading to fat accumulation and steatosis). It is precisely these adverse effects that prevented the widespread use of lomitapide, a selective MTP inhibitor. In a dose-escalation study examining the safety, tolerability, and effects on lipid levels, lomitapide reduced LDL-C by 50.9% and apoB by 55.6% when comparing levels after 4 weeks of use with the baseline levels [171].

A phase 3 study evaluated the mean reduction in LDL-C after initiating lomitapide use in patients with HoFH and observed a 50% reduction in LDL-C in 26 weeks, progressively decreasing (at a lower mean percentage) until week 78 [172]. The primary adverse effects were gastrointestinal symptoms, including abdominal cramps, nausea, vomiting, and diarrhea.

However, there are no randomized clinical trials that have evaluated the CV effects of lomitapide.

## 15. Vaccines against PCSK9

Vaccination to generate antibodies against self-antigens has proven effective in several diseases, such as cancer and hypertension. Vaccines against PCSK9 stimulate the immune system to produce specific antibodies against the PCSK9 protein. These antibodies are designed to bind to PCSK9 and neutralize it, preventing it from performing its normal function of regulating cholesterol levels. When PCSK9 is neutralized by antibodies produced through vaccination, the quantity of PCSK9 available to bind to LDL receptors on cells is reduced. This leads to an increase in the number of receptors available to remove LDL-C from the bloodstream, contributing to the reduction of circulating LDL-C levels.

Preclinical studies have demonstrated the beneficial effects of inducing anti-PCSK9 antibody production through special epitope/peptide vaccines. However, clinical studies on this topic are still limited.

Several experimental studies with positive results have been published. Landlinger et al. showed that the peptide-based AT04A vaccine promoted a sustained reduction in PCSK9 levels (≥49%) over 12 weeks [173].

L-IFPTA+ (immunogenic fused PCSK9-tetanus) is a vaccine that incorporates an immunogenic peptide onto the surface of negatively charged nanoliposomes along with the adjuvant Alhydrogel^®^. Momtazi-Borojeni et al. showed that injections of L-IFPTA+ decreased PCSK9 plasma concentrations by up to 58.5%. After 8 weeks, LDL-C and TC levels were reduced by 51.7% and 44.7%, respectively, compared to controls [174].

Other studies have also shown success in lowering LDL-C levels through the use of PCSK9 vaccines [175,176,177,178].

Momtazi-Borojeni et al. showed that nanoliposomal anti-PCSK9 vaccines induced safe, durable, and functional PCSK9-specific antibodies in hypercholesterolemic C57BL/6 mice [179].

Recently, Fowler et al. evaluated the effectiveness of virus-like particle (VLP)-based vaccines targeting epitopes in the LDL-R domain of PCSK9 [180]. VLP vaccines reduced LDL-C levels in combination with statins, whereas immunization with bivalent vaccines reduced LDL-C levels without requiring statin.

This approach aims to offer long-term control over cholesterol levels, potentially decreasing the need for frequent treatments. However, the development of these vaccines requires extensive studies to ensure their safety, effectiveness, and long-term impact on patients’ health.

## 16. Plasmapheresis

In 1980, the long-term effects of plasmapheresis for the treatment of hypercholesterolemia in two children were first reported [181]. Since then, several clinical trials have demonstrated the effectiveness and safety of the method, especially in children and patients with hypertriglyceridemia.

The most commonly used apheresis methods for removing lipoproteins from the blood are (1) double filtration; (2) LDL-C adsorption due to binding based on immunoaffinity; (3) LDL-C adsorption due to binding based on immunoaffinity; (4) heparin-induced precipitation; and (5) direct adsorption of lipoproteins (DALI) [182].

Recently, a retrospective analysis by Albayrak et al. showed that double filtration plasmapheresis reduced the levels of Lp(a), total cholesterol, LDL-C, HDL-C, and TGs in patients with FH [183].

Plasmapheresis is an effective method for removing TGs in patients with severe hypertriglyceridemia. However, the studies evaluating the safety of the method, especially in patients with hypertriglyceridemia-associated acute pancreatitis (HTG-AP) are small, retrospective, and often include mild cases.

Prior analysis of data collected for the PERFORM study showed that, after adjusting for confounders, plasmapheresis was not associated with the incidence and duration of organ failure but with increased intensive care unit requirements in patients with HTG-AP [184].

The ongoing Intensive Insulin Therapy vs. Plasmapheresis in the Management of Hypertriglyceridemia-Induced Acute Pancreatitis (Bi-TPAI) trial aims to evaluate whether intensive insulin therapy is non-inferior to plasmapheresis in patients with HTG-AP [185].

## 17. Targeted Nanotherapy

Recently, nanomedicine has proven to be a potent and effective therapy against several diseases, including dyslipidemia. Liposomes or lipid nanoparticles are commonly used as delivery vectors in pharmaceutical medicines [186,187]. Several studies have shown the successful delivery of therapeutic materials to specific tissues.

As statins are lipid-soluble drugs that cannot be directly injected into veins, and the long-term oral administration of statins can impair liver function, Rakshit et al. developed a stable simvastatin (STAT)-loaded liposome and confirmed its excellent ability to promote cholesterol efflux and its anti-inflammatory properties [188].

More recently, Salaheldin et al. developed a small oral nano-hepatic molecule-targeted anti-PCSK9 through a nanotechnological approach, efficiently reducing LDL-C levels by 50–90% [189].

## 18. Conclusions

Advances in cholesterol-lowering therapy, particularly in the context of FH, have underscored the significance of genetic interventions and the development of medications with specific actions. Inclisiran, for instance, with its focus on the enduring reduction in LDL-C, represents a paradigm shift, providing less frequent treatments with sustained effects.

Future perspectives suggest an increasing emphasis on personalized therapies tailored to the patient’s genetic profile. Additionally, there is a growing focus on research on finding drugs capable of targeting multiple factors for enhanced efficacy and a reduced incidence of side effects.

The integration of new genetic discoveries, technological advancements, and a profound understanding of drug mechanisms will enable the development of more effective therapies, ultimately reducing the CV risk in patients with dyslipidemia. However, further studies are necessary to validate the long-term safety and efficacy of these new therapeutic approaches.

## Figures and Tables

**Figure 1 pharmaceuticals-17-00289-f001:**
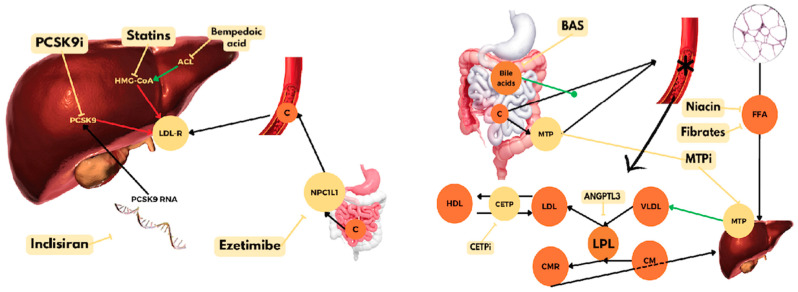
Therapeutic targets for managing dyslipidemia. Legend: ACL: ATP citrate lyase; ANGPTL3: angiopoietin-like 3; BASs: bile-acid sequestrants; CETP: cholesteryl ester transfer protein; CETPi: CETP inhibitors; CM: chylomicron; CMR: chylomicron remnants; FFA: free fatty acids; HMG-CoA: hydroxymethylglutaryl-CoA; LPL: lipoprotein lipase; MTP: microsomal triglyceride transfer protein; MTPi: MTP inhibitors; NPC1L1: Niemann–Pick C1-like 1. C: cholesterol; LDL-R: LDL receptors; VLDL: very-low-density lipoprotein.

**Figure 2 pharmaceuticals-17-00289-f002:**
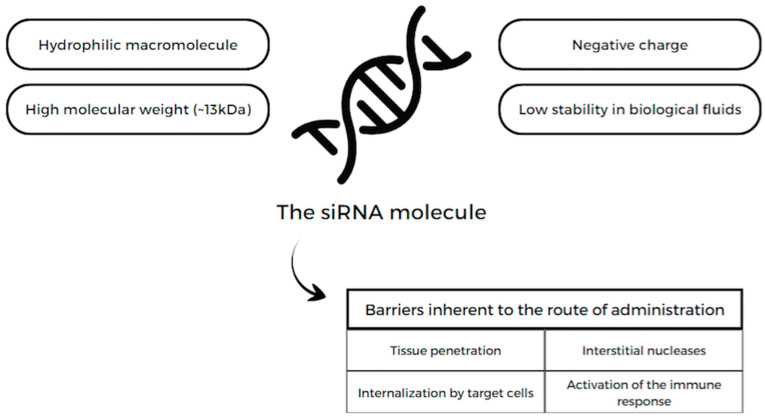
Challenges for administering siRNAs.

**Table 4 pharmaceuticals-17-00289-t004:** Main PCSK9 Inhibitors trials. Legend: ACS: acute coronary syndrome; ApoB: apolipoprotein B; CV: cardiovascular; MI: myocardial infarction; UA: unstable angina.

Study	Sample Size	Characteristics of Patients	Comparison Groups	Follow-Up	LDL-C Reduction	CV Effects
FOURIER (2017) [41]	27,564	Documented atherosclerosis + LDL-C > 70 mg/dL + on statin therapy	Evolocumab (140 mg every 2 weeks or 420 mg monthly) vs. placebo	2.2 years	59%	Reduction in CV death, MI, stroke, UA hospitalization, or coronary revascularization (HR 0.85, 95% CI 0.79 to 0.92, NNT 67)
ODYSSEY OUTCOMES (2018) [42]	18,924	ACS < 1–12 months + LDL-C > 70 mg/dL + on high-intensity statin or maximum tolerated dose	Alirocumab 75 mg vs. placebo	2.8 years	54.7%	Reduction in coronary death, MI, stroke, or UA hospitalization (HR 0.85, 95% CI 0.78 to 0.93, NNT 63)

**Table 5 pharmaceuticals-17-00289-t005:** Major clinical trials of ezetimibe. Legend: CAD: coronary artery disease; CABG: coronary artery bypass graft; CV: cardiovascular; DM: diabetes mellitus; ER-niacin: extended-release niacin; FH: familial hypercholesterolemia; MI: myocardial infarction; NA: not evaluated; NCEP ATP III: National Cholesterol Education Program Adult Treatment Panel III; PAD: peripheral artery disease; PCI: percutaneous coronary intervention; SCD: sudden cardiac death; TC: total cholesterol; TGs: triglycerides.

Study	Sample Size	Characteristics of Patients	Comparison Groups	Follow-Up	LDL-C Reduction	CV Effects
EASE (2005) [53]	3030	LDL-C > NCEP ATP III goals + TGs < 350 mg/dL + on statin	Ezetimibe 10 mg vs. placebo	6 weeks	25.8%	NE
ENHANCE (2008) [54]	720	FH	Simvastatin 80 mg vs. simvastatin 80 mg + ezetimibe 10 mg	2 years	55.6%	Reduction in carotid artery intima-media thickness
IMPROVE-IT (2015) [55]	18,144	ACS < 10 days + LDL-C 50–125 mg/dL (50–100 mg/dL if prior lipid-lowering therapy)	Simvastatin 40 mg vs. simvastatin 40 mg + ezetimibe 10 mg	6 years	25.7% vs. 43.4%	Reduction in CV events (HR 0.93, 95% CI 0.89 to 0.99, NNT 50)
HIJ-PROPER (2017) [56]	1734	ACS + LDL-C > 100 mg/dL + TGs < 400 mg/dL	Pitavastatin (1–4 mg) vs. pitavastatin (1–4 mg) + ezetimibe 10 mg	3.8 years	37.6% vs. 51.7%	No significant differences in CV events
EWTOPIA 75 (2019) [57]	3411	≥75 years + LDL-C > 140 mg/dL + ≥1 high risk factor	Ezetimibe 10 mg vs. control	4.1 years	25.9%	Reduction in SCD, MI, PCI, or CABG, or stroke (HR 0.66 95% CI 0.50 to 0.86, NNT 38.5)
RACING (2023) [58]	3780	Documented atherosclerotic disease	Rosuvastatin 10 mg + ezetimibe 10 mg vs. rosuvastatin 20 mg	3 years	58% vs. 72%	No significant differences in CV events

**Table 6 pharmaceuticals-17-00289-t006:** Major clinical trials of niacin. Legend: ACS: acute coronary syndrome; CAD: coronary artery disease; CABG: coronary artery bypass graft; CV: cardiovascular; DM: diabetes mellitus; ER-niacin: extended-release niacin; MI: myocardial infarction; PAD: peripheral artery disease; TC: total cholesterol; TGs: triglycerides; TIA: transient ischemic attack.

Study	Sample Size	Characteristics of Patients	Comparison Groups	Follow-Up	Lipids Effect	CV Effects
CDP (1975) [81]	8341	Men with previous MI	Niacin (1 g 3×) vs. placebo	6 years	Reduction TC 9.9%	Reduction in nonfatal MI 27% Reduction in cerebrovascular events 24%
Stockholm trial (1977) [82]	558	ACS + aged <70 years	Clofibrate (1 g 2×) + niacin (1 g 3×) vs. placebo	5 years	Reduction TC 26%	Reduction in CV events (HR 0.59, 95% CI 0.41 to 0.84)
CLAS (1987) [83]	162	Men aged 40–59 years + previous CABG	Niacin (3–12 g) + colestipol (30 g) vs. placebo	2 years	Reduction LDL-C 43% Increase HDL-C 37%	Reduction in atherosclerotic regression (16.2% vs. 2.4%, *p* = 0.002)
HATS (2001) [84]	160	CAD + LDL-C < 140 mg/dL + TGs < 400 mg/dL	Simvastatin (10–20 mg) + niacin (250–1000 mg 2×) vs. antioxidant vs. simvastatin + niacin + antioxidant vs. placebo	3 years	Reduction LDL-C 42% Increase HDL-C 26%	Reduction in death, MI, stroke, or revascularization
ARBITER-2 (2004) [85]	149	CAD + on statin therapy	ER-niacin (1000 mg 1×) vs. placebo	1 year	Increase HDL-C 21%	Reduction in progression of carotid intima-media thickness
ARBITER-6 (2009) [86]	208	CAD or CAD risk equivalent on statin therapy	ER-niacin (2 g 1×) vs. ezetimibe (10 mg)	1.2 years	Reduction LDL-C (20% vs. 12%) Increase HDL-C (+18% vs. −7%)	Reduction in progression of carotid intima-media thickness
AFREGS (2005) [87]	143	CAD + LDL-C < 160 mg/dL + HDL-C < 40 mg/dL	Niacin (240–3000 mg) + gemfibrozil (600 mg 2×) + cholestyramine (2–16 g 1×) vs. placebo	2.5 years	Reduction LDL-C 24% Increase HDL-C 38%	Reduction in angiographic progression (34.72% vs. 36.02%, *p* = 0.0002)
AIM-HIGH (2011) [71]	3414	Aged >45 years + CV disease	ER-niacin (1.5–2 g 1×) vs. placebo	3 years	Reduction LDL-C 14% Increase HDL-C 25%	No significant differences in CV events
HPS2-THRIVE (2014) [88]	25,673	Aged >50 years + CV disease	ER-niacin (2 g 1×)/laropiprant (40 mg) vs. placebo	3.9 years	Reduction LDL-C 10% Increase HDL-C 6%	No significant differences in CV events

**Table 7 pharmaceuticals-17-00289-t007:** Major clinical trials and epidemiological studies of fibrates. Legend: CAD: coronary artery disease; CV: cardiovascular; MI: myocardial infarction; NE: not evaluated; PAD: peripheral artery disease; TGs: triglycerides; T2DM: type 2 diabetes mellitus.

Study	Sample Size	Characteristics of Patients	Comparison Groups	Follow-Up	Lipids Effect	CV Effects
HHS (1987) [91]	4081	Men with non-HDL-C > 200 mg/dL	Gemfibrozil (1200 mg) vs. placebo	5 years	Increase HDL-C 11% Reduction LDL-C 11% Reduction TGs 35%	Reduction in incidence of CAD
VA-HIT (1999) [92]	2531	Documented CAD + LDL < 140 mg/dL	Gemfibrozil (1200 mg) vs. placebo	5.1 years	Reduction TGs 31% Increase HDL-C 6%	Reduction in MI or coronary death
BIP (2000) [93]	3090	Previous MI or stable angina + LDL-C < 180 mg/dL	Bezafibrate (400 mg) vs. placebo	6.2 years	Reduction TGs 21% Increase HDL-C 18%	No significant differences in CV events
LEADER (2002) [94]	1568	Men with lower PAD	Bezafibrate (400 mg) vs. placebo	4.6 years	NE	No significant differences in CV events
FIELD (2005) [95]	9795	T2DM	Fenofibrate (200 mg) vs. placebo	5 years	Reduction TGs 29% Increase HDL-C 5% Reduction LDL-C 12%	No significant differences in CV events
ACCORD-Lipid (2010) [96]	5518	T2DM + CV risk factor or documented CV disease	Fenofibrate (160 mg) vs. placebo	4.7 years	Increase HDL-C 9% Reduction TGs 23%	No significant differences in CV events
ECLIPSE-REAL (2019) [97]	10,705	Metabolic syndrome	Statin vs. statin + fenofibrate	6 years	Reduction TGs with statin + fenofibrate	Reduction in CAD, stroke, or CV death with statin + fenofibrate (HR 0.74, 95% CI 0.58 to 0.93)
PROMINENT (2022) [98]	10,497	T2DM + TGs 200–499 mg/dL + HDL < 40 mg/dL	Pemafibrate (0.4 mg) vs. placebo	3.4 years	Reduction TGs 31.1%	No significant differences in CV events

**Table 9 pharmaceuticals-17-00289-t009:** Characteristics of the main published trials involving CETP inhibitors. Legend: ACS: acute coronary syndrome; CAD: coronary artery disease; CV: cardiovascular; HeFH: heterozygous familial hypercholesterolemia; MI: myocardial infarction; NE: not evaluated.

Study	Sample Size	Characteristics of Patients	Comparison Groups	Follow-Up	Lipids Effect	CV Effects
ILLUMINATE (2007) [111]	15,067	CV disease	Torcetrapib (60 mg/d) vs. placebo	1 year	Increase HDL-C 72.1% Reduction LDL-C 24.9%	Reduction in CV events (HR 1.25, 95% CI 1.09 to 1.44)
dal-OUTCOMES (2012) [112]	15,871	ACS	Dalcetrapib (600 mg/d) vs. placebo	31 months	Increase HDL-C 30%	No significant differences in CV events
ACCELERATE (2017) [113]	12,092	CV disease	Evacetrapib (130 mg/d) vs. placebo	26 months	Increase HDL-C 133.2% Reduction LDL-C 31.1%	No significant differences in CV events
REVEAL (2017) [114]	30,449	Atherosclerotic disease + on atorvastatin therapy + mean LDL-C 61 mg/dL	Anacetrapib (100 mg/d) vs. placebo	4.1 years	Reduction non-HDL-C 18%	Reduction in coronary death, MI, or coronary revascularization (HR 0.91, 95% CI 0.85 to 0.97, NNT 100)
ROSE (2022) [115]	120	Dyslipidemia on statin treatment	Obicetrapib (10 mg/d) vs. placebo	8 weeks	Reduction LDL-C 51%	NE

**Table 10 pharmaceuticals-17-00289-t010:** Characteristics of the main published trials involving inclisiran. Legend: ASCVD: atherosclerotic cardiovascular disease; CV: cardiovascular; NE: not evaluated.

Study	Sample Size	Characteristics of Patients	Comparison Groups	Follow-Up	LDL-C Reduction	CV Effects
ORION-1 (2017) [124]	501	Patients at high risk for CV disease and elevated LDL-C	Different doses of inclisiran vs. placebo	180 days	27.9% (200 mg inclisiran); 38.4% (300 mg inclisiran); 41.9% (500 mg inclisiran); 35.5% (double dose 100 mg inclisiran); 44.9% (double dose 200 mg inclisiran); 52.6% (double dose 300 mg inclisiran)	NE
ORION-3 (2023) [125]	382	Patients at high risk for CV disease and elevated LDL-C	Inclisiran-only (patients who already received inclisiran continued to receive it) vs. switching-arm (patients who received placebo in ORION-1 first received evolocumab for 1 year and then switched to inclisiran)	4 years	44.2%	NE
ORION-5 (2023) [126]	56	HoFH	Inclisiran (300 mg) vs. placebo	150 days	1.68%	NE
ORION-9 (2020) [127]	482	HeFH	Inclisiran (300 mg) vs. placebo	510 days	39.7%	NE
ORION-10 and ORION-11 (2020) [128]	1561 (ORION-10) and 1617 (ORION-11)	ASCVD (ORION-10 trial) and ASCVD-risk equivalent (ORION-11 trial) and elevated LDL-C on maximum tolerated dose of statin	Inclisiran (284 mg) vs. placebo	510 days	52.3% (ORION-10) 49.9% (ORION-11)	NE

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
