# Peer review of "Dyslipidemia: A Narrative Review on Pharmacotherapy"

_pharmaceuticals, 2024, doi:10.3390/ph17030289_

Round 1
Reviewer 1 Report
Comments and Suggestions for Authors
This paper, submitted by Herling de Oliveira et al, provides a review of cholesterol-lowering drugs, providing a comprehensive overview of the mechanisms, effects, safety, and clinical applications of different types of lipid-lowering drugs. As a review study, the authors provide an extensive and informative overview of relevant studies. However there are some recommendations and questions:
1. The introduction of the article should present information about the epidemiology, harms, and treatment of dyslipidemia to provide context for what follows.
2. The methods section of this article should detail the article's inclusion and exclusion criteria, and the methods used for data extraction and analysis, to add credibility to the article.
3. Most of reviews have published about these drugs, I strong recommend to revised as an update review, and remove the old drugs such as statin and ezetimibe parts if there are no update.
4. Emphasizing the pharmacological mechanism and role of new lipid-lowering drugs maybe intesting to the readers.
Author Response
Reviewer 1
1. The introduction of the article should present information about the epidemiology, harms, and treatment of dyslipidemia to provide context for what follows.
A. Thanks for the suggestion. The focus of our manuscript is to review the pharmacological management of hypercholesterolemia. In this sense, the abstract already condenses the objectives of the review.
2. The methods section of this article should detail the article's inclusion and exclusion criteria, and the methods used for data extraction and analysis, to add credibility to the article.
A. Thanks for the observation. As this paper is a narrative review, the most relevant articles on dyslipidemia were selected from the aforementioned databases.
3. Most of reviews have published about these drugs, I strong recommend to revised as an update review, and remove the old drugs such as statin and ezetimibe parts if there are no update.
A. Thanks. We chose to review pharmacological therapies for dyslipidemia and not just an update on pharmacological treatment.
4. Emphasizing the pharmacological mechanism and role of new lipid-lowering drugs maybe interesting to the readers.
A. Thanks. I totally agree with you. In all sections with the drugs mentioned, we detail the mechanism of action of the drugs. Additionally, we associate 2 figures with the manuscript.
Reviewer 2 Report
Comments and Suggestions for Authors
Dear authors, in this manuscript you try to synthesize the vast evidence about the pharmacological interventions in hyperlipidemic diseases, however there are some recommendations that you could addressed to increase the relevance of your manuscript. In the next lines you can find some recommendations:
1) The abstract section is too poor. Please add more information about the findings in the review process and some futures directions in the drugs development. Also could be great if you can add information about the role of economic factors to access to the new drugs which will be more expensive and not all the health systems around the world could afford it.
2) In the introduction section please explain the epidemiological significance of this disease, the prevalence, the cost of the disease, the role of other intervention such as a healthy diet, physical activity, exercise, and political interventions to control the disease. At the end of the introduction you can add information about the increase needed of drugs to control the disease and the aim of this review.
3) Methods. I understand this is a “narrative review” however, the authors only search for specific drugs and probably this could induces some bias in the evidences collected. For that reason it is necessary that the authors explain why they did the search in that way. Can you justify it? For example, if you selected the drugs because are the more indicated to patients or in the guidelines? Also, the figure 1 is not on the manuscript submitted.
4) In the results sections looks like the most important drugs is statins because the authors have divided the evidence about them in primary and secondary prevention but the others drugs are not commented in the same way. Can the authors explain why?
5) There are some molecules that the authors don’t mentioned in this manuscript, for example: permafibrate. Also will be very interesting if the authors can add the date of approval of each new molecules. Also, will be great if the authors can add information about new molecules in development for example doing a search in clinical trials. You can review this work: https://www.medigraphic.com/pdfs/cardiovascuar/cms-2021/cmss213s.pdf
6) At least, can the authors add information about the clinical doses recommended for each type of patients?
Author Response
Reviewer 2
Dear authors, in this manuscript you try to synthesize the vast evidence about the pharmacological interventions in hyperlipidemic diseases, however there are some recommendations that you could addressed to increase the relevance of your manuscript. In the next lines you can find some recommendations:
1) The abstract section is too poor. Please add more information about the findings in the review process and some futures directions in the drugs development. Also could be great if you can add information about the role of economic factors to access to the new drugs which will be more expensive and not all the health systems around the world could afford it.
A. Thanks for the suggestion. The focus of our manuscript is to review the pharmacological management of hypercholesterolemia. In this sense, the abstract already condenses the objectives of the review. Anyway, we added the following text to the abstract:
“However, these new drugs such as PCSK9 inhibitors still have a high cost. This narrative review summarizes current and emerging therapies for the management of patients with dyslipidemia.”
2) In the introduction section please explain the epidemiological significance of this disease, the prevalence, the cost of the disease, the role of other intervention such as a healthy diet, physical activity, exercise, and political interventions to control the disease. At the end of the introduction you can add information about the increase needed of drugs to control the disease and the aim of this review.
A. Thanks for the suggestion. Our review focuses only on the pharmacological management of hypercholesterolemia. Adding information on epidemiological aspects, costs and other non-pharmacological interventions is beyond the scope of this review. Furthermore, at the end of the introduction, you can read the objectives of the review: "The aim of this article was to review the currently available therapies and emerging therapeutic agents for the management of patients with dyslipidemia, in light of recent evidence and guideline recommendations."
3) Methods. I understand this is a “narrative review” however, the authors only search for specific drugs and probably this could induces some bias in the evidences collected. For that reason it is necessary that the authors explain why they did the search in that way. Can you justify it? For example, if you selected the drugs because are the more indicated to patients or in the guidelines? Also, the figure 1 is not on the manuscript submitted.
A. Important observation. We chose to select the medications most commonly used in clinical practice, recognized by evidence-based medicine and recommended by guidelines. We attach figure 1 to the manuscript. There probably must have been some error in the manuscript submission process.
4) In the results sections looks like the most important drugs is statins because the authors have divided the evidence about them in primary and secondary prevention but the others drugs are not commented in the same way. Can the authors explain why?
A. Several clinical trials on statins in primary and secondary prevention have been published in recent decades. The same did not occur with other classes of medications. Therefore, we decided to divide the management of hypercholesterolemia with statins into these two scenarios: primary and secondary prevention.
5) There are some molecules that the authors don’t mentioned in this manuscript, for example: permafibrate. Also will be very interesting if the authors can add the date of approval of each new molecules. Also, will be great if the authors can add information about new molecules in development for example doing a search in clinical trials. You can review this work: https://www.medigraphic.com/pdfs/cardiovascuar/cms-2021/cmss213s.pdf
A. Thank you very much. We discuss the PROMINENT trial and the medication pemafibrate. You can find this information on page 27 of the manuscript. Additionally, we quote and comment on several modern medications and gene therapy. Following your suggestion, we analyzed the recommended article (https://www.medigraphic.com/pdfs/cardiovascuar/cms-2021/cmss213s.pdf). Innovative molecules, such as lomitapide, mipomersen, pemafibrate, inclisiran and bempedoic acid were cited and analyzed by us in the manuscript.
​
6) At least, can the authors add information about the clinical doses recommended for each type of patients?
A. Thank you for the suggestion. Medication doses can be easily found in the manuscript tables.
Reviewer 3 Report
Comments and Suggestions for Authors
The manuscript by Lucas Lentini Herling de Oliveira and co-authors compiles significant insights on dyslipidemias.
Nonetheless, several aspects need refinement to enhance its overall quality.
1. The article title is too general. It should specify the fact that it is not a therapeutic guideline and rather an overview of clinical studies and emerging therapies in dyslipidemia.
2. The introduction is short and is based on a single reference from the literature. The authors must expand this section based on more references.
3. In the method section, the authors mentioned figure 1, but no figure appears in the article.
4. Table 9 should rather be a figure or a diagram.
5. An abbreviation list would be useful as there are many abbreviation in the text.
I believe that after this revision is made, the article will become suitable for publication.
Comments on the Quality of English LanguageMinor editing of English language required.
Author Response
Reviewer 3
The manuscript by Lucas Lentini Herling de Oliveira and co-authors compiles significant insights on dyslipidemias.
Nonetheless, several aspects need refinement to enhance its overall quality.
1. The article title is too general. It should specify the fact that it is not a therapeutic guideline and rather an overview of clinical studies and emerging therapies in dyslipidemia.
A. Thank you for your observation. We changed the title to "Dyslipidemia: A narrative Review on Pharmacotherapy"
2. The introduction is short and is based on a single reference from the literature. The authors must expand this section based on more references.
A. Thanks for the sugestion. We have added the following text to the introduction:
" Although statins remain the first line of pharmacotherapy, there are novel lipid lowering therapies currently available for use, such as PCSK9 inhibitors, gene therapy, including small interfering RNAs (inclisiran), ANGPTL3 inhibitors (evinacumab), CRISPR/Cas9 , antisense Oligonucleotides (mipomersen), apoB and MTP Inhibitors, and, finally, vaccines against PCSK9."
3. In the method section, the authors mentioned figure 1, but no figure appears in the article.
A. Thank you. We attach figure 1 to the manuscript. There probably must have been some error in the manuscript submission process.
4. Table 9 should rather be a figure or a diagram.
A. Thank you very much. We replace table 9 with a figure.
5. An abbreviation list would be useful as there are many abbreviation in the text.
A. Thank you. We have added an abbreviation list in the end of the manuscript.
Round 2
Reviewer 1 Report
Comments and Suggestions for Authors
Author Response
I do not identify reviewer comments
Reviewer 2 Report
Comments and Suggestions for Authors
Dear authors, thanks for the improvement in your manuscript. However, there are still missing some recommendations:
1) Please review the entire manuscript to avoid some mistakes/errors in the English grammar.
2) The abstract section is still poorly, please add information about the conclusion of your review, the number of drugs available, new or prospective molecules, and the principal results or expectations about the drugs.
3) Introduction section. The solicitude to add information about the epidemiologic situation, related factors and more is to contextually the need to use drug therapy and still do research to find new molecules.
4) Methods section. Please add in this section that you review the drugs that are more recommended in the guidelines.
Author Response
We thank the reviewer for the suggestions and observations. We believe that they made the paper more informative and complete.
1) Please review the entire manuscript to avoid some mistakes/errors in the English grammar.
A. Thank you so much. We submit the paper for English review and correct grammatical errors.
2) The abstract section is still poorly, please add information about the conclusion of your review, the number of drugs available, new or prospective molecules, and the main results or expectations about the drugs.
A. Thanks for the observation. We insert the following information into the abstract:
"An emerging field in dyslipidemia pharmacotherapy is research in genetic therapies and genetic modulation. Understanding the genetic mechanisms underlying lipid alterations may lead to the development of personalized treatments that directly target the genetic causes of dyslipidemia. RNA messenger (mRNA)-based therapies are also being explored, offering the ability to modulate gene expression to normalize lipid levels. Furthermore, nanotechnology opens new possibilities in drug delivery for treating dyslipidemia. Controlled-release systems, nanoparticles, and liposomes can enhance the effectiveness and safety of medications by providing more precision and sustained release."
3) Introduction section. The request to add information about the epidemiological situation, related factors and more is to contextually the need to use drug therapy and still do research to find new molecules.
A. Thanks for the suggestion. We insert the following information into the introduction section:
"Dyslipidemia is one of the most important risk factors for atherosclerotic cardiovascular disease (ASCVD). In 2017, high non-high-density lipoprotein cholesterol (HDL-C) was responsible for an estimated 3.9 million deaths worldwide [1]. Therefore, lipid -lowering therapies, especially statins, have been shown to be cost-effective or cost-saving, particularly in people with a high cardiovascular (CV) disease risk [2-5]. that effective community-based prevention strategies promoting lifestyle modification (e.g., dietary improvement and regular physical activity) are also needed to control dyslipidemia."
4) Methods section. Please add in this section that you review the drugs that are most recommended in the guidelines.
A. Thanks for the suggestion. We insert the following information into the Methods section:
"The primary lipid-lowering drugs recommended by the European and American dyslipidemia guidelines are included in this review."
Additionally, we considered it relevant and included a short text on nanomedicine.
" 17. Targeted nanotherapy
Recently, nanomedicine has proven to be a potent and effective therapy against several diseases, including dyslipidemia. Liposomes or lipid nanoparticles are commonly used as delivery vectors in pharmaceutical medicines [190,191]. Several studies have shown the successful delivery of therapeutic materials to specific tissues.
As statins are lipid-soluble drugs that cannot be directly injected into veins, and the long-term oral administration of statins can impair liver function, Rakshit et al. developed a stable simvastatin (STAT)-loaded liposome and confirmed its excellent ability to promote cholesterol efflux and its anti-inflammatory properties [192].
More recently, Salaheldin et al. developed a small oral nano-hepatic molecule targeted anti-PCSK9 through a nanotechnological approach, efficiently reducing LDL-C levels by 50-90% [193]."